# EXPANDING SMALL-SCALE DATASETS WITH GUIDED IMAGINATION

## ABSTRACT

The power of Deep Neural Networks (DNNs) depends heavily on the training data quantity, quality and diversity. However, in many real scenarios, it is costly and time-consuming to collect and annotate large-scale data. This has severely hindered the application of DNNs. To address this challenge, we explore a new task of *dataset expansion*, which seeks to automatically create new labeled samples to expand a small dataset. To this end, we present a Guided Imagination Framework (GIF) that leverages the recently developed big generative models (*e.g.,* DALL-E2) and reconstruction models (*e.g.,* MAE) to "imagine" and create informative new data from seed data to expand small datasets. Specifically, GIF conducts imagination by optimizing the latent features of seed data in a semantically meaningful space, which are fed into the generative models to generate photo-realistic images with *new* contents. For guiding the imagination towards creating samples useful for model training, we exploit the zero-shot recognition ability of CLIP and introduce three criteria to encourage informative sample generation, *i.e., prediction consistency*, *entropy maximization* and *diversity promotion*. With these essential criteria as guidance, GIF works well for expanding datasets in different domains, leading to 29.9% accuracy gain on average over six natural image datasets, and 10.4% accuracy gain on average over three medical image datasets.

## 1 INTRODUCTION

Having a sufficient amount of training data is crucial for unleashing the power of deep neural networks (DNNs) (Deng et al., 2009; Qi & Luo, 2020). However, in many fields, collecting large-scale datasets is expensive and time-consuming (Qi & Luo, 2020; Zhang et al., 2020), resulting in limited dataset sizes which make it difficult to fully utilize DNNs. To address this data limitation issue and reduce the cost of manual data collection/annotation, we explore *dataset expansion* in this work, which seeks to build an automatic data generation pipeline for expanding a small dataset into a larger and more informative one, as illustrated in Figure 1 (left).

There are some research attempts that could be applied to dataset expansion. Among them, data augmentation (DeVries & Taylor, 2017; Cubuk et al., 2020; Zhong et al., 2020) applies pre-defined transformations to each image for enriching datasets. However, these transformations mostly affect the surface visual characteristics of an image, but have a minimal effect on the actual image content. Therefore, the brought new information is limited, and cannot sufficiently address the limited-data issue in small datasets. Besides, some recent studies (Zhang et al., 2021c; Li et al., 2022) utilize generative adversarial networks (GANs) (Goodfellow et al., 2014; Brock et al., 2018) to synthesize images for model training. They, however, require a sufficiently large dataset for in-domain GAN training, which is not feasible in the small-data scenario. Moreover, the generated images are often not well-annotated, limiting their utility for DNN training. Therefore, both of them are unable to effectively resolve the dataset expansion problem.

For an observed object, humans can easily imagine its different variants in various shapes, colors or contexts, relying on their accumulated prior understanding of the world (Warnock & Sartre, 2013; Vyshedskiy, 2019). Such an imagination process is highly useful for dataset expansion, since it does not simply perturb the object's appearance but applies rich prior knowledge to create object variants with new information. Meanwhile, recent breakthroughs in large-scale generative models (*e.g.,* DALL-E2 (Ramesh et al., 2022)) have demonstrated that generative models can effectively capture

Figure 1: **Dataset expansion** aims to create data with *new* information to enrich small datasets for training DNN models better (left). The ResNet50 trained on the expanded datasets by our proposed method performs much better than the one trained on the original small datasets (right).

the sample distribution of extremely large datasets (Schuhmann et al., 2021; Byeon et al., 2022) and show encouraging abilities in generating photo-realistic images with a rich variety of contents. This motivates us to explore their capabilities as prior models to develop a computational data imagination pipeline for dataset expansion, by imagining different sample variants from seed data. However, deploying big generative models for dataset expansion is highly non-trivial, complicated by several key challenges, including how to generate samples with correct labels, and how to make sure the created samples are useful for model training.

To handle these challenges, we conduct a series of studies (cf. Section 3), from which we make two important findings. First, the CLIP model (Radford et al., 2021), which offers excellent zero-shot classification abilities, can map latent features of category-agnostic generative models to the specific label space of the target small dataset. This is helpful for generating samples with correct labels. Second, we empirically find three informativeness criteria crucial for generating effective training data: 1) *zero-shot prediction consistency* to ensure that the imagined image is class-consistent with the seed image; 2) *entropy maximization* to encourage the imagined images to bring more information; 3) *diversity promotion* to encourage the imagined images to have diversified contents.

In light of the above findings, we propose the Guided Imagination Framework (GIF) for dataset expansion. Specifically, given a seed image, GIF first extracts its latent feature with the prior generative model. Different from data augmentation that imposes variation over the raw image, GIF optimizes the variation over the latent feature. Thanks to the guidance carefully designed by our discovered criteria, the latent feature is optimized to provide more information while maintaining its class semantics. This enables GIF to create informative new samples, with class-consistent semantics yet higher content diversity, to expand small datasets for model training. Considering that DALL-E2 have been shown to be powerful in generating images and MAE (He et al., 2022) is excellent at reconstructing images, we explore their use as prior models for imagination in this work.

We evaluate the proposed method on both small-scale natural and medical image datasets. As shown in Figure 1 (right), compared to the ResNet50 model trained on the original dataset, our method improves the model performance by a large margin across a variety of visual tasks, including fine-grained object classification, texture classification, cancer pathology detection, and ultrasound image classification. More specifically, GIF obtains 29.9% accuracy gain on average over six natural image datasets, and 10.4% accuracy gain on average over three medical image datasets. Moreover, we show that the expansion efficiency of our method is much higher than expansion with existing augmentation methods. For example, 5× expansion by our GIF-DALLE method already outperforms 20× expansion by Cutout, GridMask and RandAugment on the Cars and DTD datasets. In addition, the expanded datasets can be directly used to train different model architectures (*e.g.,* ResNeXt, WideResNet and MobileNet), leading to consistent performance improvement.

## 2 RELATED WORK

**Learning with synthetic images.** Training models with synthetic images is a promising direction (Jahanian et al., 2022). DatasetGANs (Zhang et al., 2021c; Li et al., 2022) explore GAN models (Isola et al., 2017; Esser et al., 2021) to generate images for segmentation model training. However, as the generated images are without labels, they need manual annotations on generated images to train a label generator for annotating synthetic images. In contrast, our dataset expansion aims to expand a real small dataset to a larger labeled one in a fully automatic manner, without involving

human annotators. In this work, we did not explore designing more advanced generative models as this is not our focus, and we leave the discussion on image synthesis to Appendix A for reference.

**Data augmentation.** Data augmentation has been widely used to improve the generalization of DNNs (Shorten & Khoshgoftaar, 2019), which typically generates new images with manually specified rules, *e.g.*, image manipulation (Yang et al., 2022), image erasing (DeVries & Taylor, 2017; Zhong et al., 2020), image mixup (Zhang et al., 2021a; Hendrycks et al., 2019), and learning to select from a set of transformations for augmentation (Cubuk et al., 2019; 2020). Despite their effectiveness in some applications, most augmentation methods apply pre-defined transformations to enrich datasets, which only locally varies the pixel values of images and cannot generate images with highly diversified contents. Moreover, most methods cannot guarantee the effectiveness of the augmented samples for model training. As a result, they cannot effectively address the issue of lack of information in small datasets, and their efficiency of dataset expansion is low. In comparison, our GIF framework leverages powerful generative models trained on large datasets and guides them to generate more informative and diversified images, and thus can expand datasets more efficiently. More discussions on data augmentation are provided in Appendix A.

## 3    PROBLEM STATEMENT AND PRELIMINARY STUDIES

**Problem statement**. To address the common data scarcity challenge when deploying DNN models, this paper explores a new task, *dataset expansion*. Without loss of generality, we consider image classification problems. We are given a small-scale image dataset $\mathcal{D}_o = \{x_i, y_i\}_{i=1}^{n_o}$, where $n_o$ denotes the number of samples, and $x_i$ denotes an instance with class label $y_i$. Dataset expansion aims to generate a set of new synthetic samples $\mathcal{D}_s = \{x'_j, y'_j\}_{j=1}^{n_s}$ to enlarge the original dataset, such that a DNN model trained on the expanded dataset $\mathcal{D}_o \cup \mathcal{D}_s$ outperforms the model trained on $\mathcal{D}_o$ significantly. The key is that the synthetic sample set $\mathcal{D}_s$ should be highly-related to the original dataset $\mathcal{D}_o$ while *bringing sufficient new information* beneficial for model training.

### 3.1    A PROPOSAL FOR COMPUTATIONAL IMAGINATION MODELS

Given an observed object, humans can easily imagine its different variants, like the object in various colors, shapes or contexts, based on their accumulated prior knowledge about the world (Warnock & Sartre, 2013; Vyshedskiy, 2019). Inspired by this, we attempt to build a computational model to simulate this imagination process for dataset expansion. It is known that deep generative models are trained to capture the full distribution of a training dataset and can well maintain its distribution knowledge. We can query a well-trained generative model to generate new samples with similar characteristics presented by its training dataset. More crucially, recent deep generative models (*e.g.*, DALL-E2) have shown impressive abilities in capturing the sample distribution of extremely large datasets and generating photo-realistic images with various contents, which inspires us to explore them as the prior model to build the data imagination pipeline.

Specifically, given a pre-trained generative model $G$, and a seed example $(x, y)$ from the small dataset to expand, we formulate the imagination of $x'$ from $x$ as $x' = G(f(x) + \delta)$. Here $f(\cdot)$ is an image encoder of the generative model to transform the raw image into an embedding for imagination with the generative model, and $\delta$ is a perturbation applied to $f(x)$ such that $G$ can generate $x'$ different from $x$. A simple choice of $\delta$ would be a random noise sampled from a Gaussian distribution. We will discuss how to optimize $\delta$ to provide useful guidance in the following section.

Dataset expansion requires the created samples to have correct category labels for model training. However, ensuring the generated $x'$ to have the same label $y$ as $x$ is highly non-trivial, because it is hard to maintain the class semantics of the seed sample in the embedding space of $f(x)$ after perturbation and the pre-trained generative models are usually category-agnostic for the target dataset. We resort to CLIP (Radford et al., 2021) to address this issue. CLIP is trained by contrastive learning on a large-scale image-text dataset, such that its image encoder can project images to an embedding space well aligned with a rich natural language corpus (Radford et al., 2021; Tian et al., 2022; Wortsman et al., 2022). We propose to leverage the CLIP image encoder in our computational imagination model to map any sample $x$ from the target dataset to the embedding aligned with the embedding of its category name $y$, and thus it is convenient to take the embedding of label $y$ as a reference anchor to regularize the image embedding $f(x)$ to avoid changing its class.

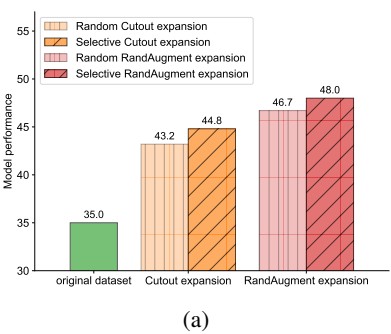 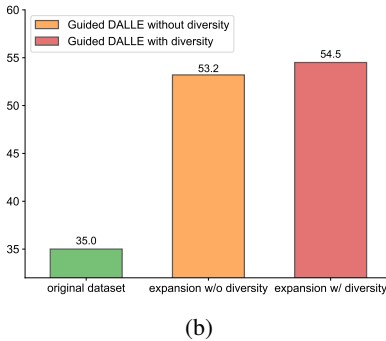

(a)                                              (b)

Figure 2: **Effectiveness of the informativeness criteria for sample creation**. (a) Comparison between random expansion and our selective expansion on CIFAR100-Subset. (b) Comparison between the guided DALLE expansion with and without diversity promotion on CIFAR100-Subset.

It is worth noting that in this work we do not seek to develop a biologically plausible imagination model that exactly follows true human brain dynamics and rules. Instead, we draw inspiration from the imagination activities of human brains and propose a pipeline to leverage the well pre-trained big generative models to explore dataset expansion.

### 3.2 How to guide the imagination for effective dataset expansion?

The proposed data imagination pipeline leverages big generative models to create new samples from the seeds to expand datasets. However, it is unclear what kind of samples are effective and helpful for model training. To determine this question and optimize $\delta$ accordingly in the above pipeline, we conduct several preliminary studies and make the following observations.

To guide generative models to create data that are useful for model training, the sample should have the same class semantics (without label change from the seed sample) but bring new information. To achieve these properties, we leverage the zero-shot prediction ability of CLIP and design three informativeness criteria based on the CLIP-mapped label space: *zero-shot prediction consistency*, *entropy maximization* and *diversity promotion*. The criterion of prediction consistency means that the zero-shot predictions for the synthetic image and the seed image should be the same, which ensures that the class semantics of the imagined image is consistent with that of the seed one. Entropy maximization encourages the synthetic image to have larger zero-shot prediction entropy than the seed image, and thus it is more challenging for classification and brings new information for model training. Moreover, diversity promotion encourages the synthetic images for the same seed image to be as diversified as possible so that they can bring more diverse information.

**Class consistency and entropy maximization**. We first start with exploratory experiments on a subset of CIFAR100 (Krizhevsky et al., 2009) to pinpoint whether achieving the criteria of class consistency and entropy maximization leads to more informative samples. Here, CIFAR100-subset is built for simulating small-scale datasets by randomly sampling 100 instances per class from the original CIFAR100, and the total sample number is 10,000. We synthesize samples based on existing data augmentation methods (*i.e.,* RandAugment and Cutout (DeVries & Taylor, 2017)) and expand CIFAR100-subset by 5×. Meanwhile, we conduct selective augmentation expansion by selecting the samples with the same zero-shot prediction but higher prediction entropy compared to their seed samples, until we reach the required expansion ratio per seed sample. The goal is to examine whether the selective synthetic samples are more useful for model training. As shown in Figure 2a, the selective expansion strategy outperforms random expansion by 1.3% to 1.6%, meaning that the selected samples are more informative for model training. Compared to random augmentation, selective expansion can filter out the synthetic samples with lower prediction entropy and those with higher entropy but different predictions. The remaining samples thus contain more information while preserving the same class semantics, leading to better expansion effectiveness.

**Diversity promotion.** To avoid "imagination collapse" in which the generative models generate excessively similar or duplicate samples, we further introduce the criterion of diversity promotion. To study its effectiveness, we resort to a powerful generative model (*i.e.,* DALL-E2) as the prior model to generate images and expand CIFAR100-subset by 5×, where the guided expansion scheme and the implementation of diversity promotion will be introduced in the following section. As shown in Figure 2b, diversity promotion can further improve accuracy by 1.3%, demonstrating that it can bring more diverse information to boost model training.

Figure 3: Overview of our proposed GIF-DALLE method, which expands small datasets by creating informative new samples with guided imagination. Here, we resort to DALL-E2 as the prior generative model, in which the image/text encoders are CLIP image/text encoders while the decoder is the diffusion model of DALL-E2. Moreover, $\oplus$ denotes guided residual multiplicative perturbation.

## 4 GIF: A GUIDED IMAGINATION FRAMEWORK FOR DATASET EXPANSION

In light of the above studies, we propose a simple GIF Framework for dataset expansion, by guiding the imagination method built on big pre-trained generative models via the aforementioned criteria.

Given a seed image $x$ from a target small dataset, we first extract its latent feature $f(x)$ based on the encoder of a pre-trained model (*e.g.,* the CLIP image encoder $f_{\text{CLIP-I}}$). Different from data augmentation that imposes variation over the raw RGB images, our proposed pipeline explores optimizing the variation over the sample latent features. Thanks to the careful method designed with the criteria investigated above, the varied latent features are able to maintain the sample class semantics while providing more new information for model training.

To detail our proposed framework, we utilize DALL-E2 as the prior generative model for illustration. As shown in Figure 3, DALL-E2 is built by adopting CLIP image/text encoders $f_{\text{CLIP-I}}$ and $f_{\text{CLIP-T}}$ as its image/text encoders and using a pre-trained diffusion model $G$ as its decoder. To create a set of new images $x'$ from the seed image $x$, GIF first repeats its latent feature $f = f_{\text{CLIP-I}}(x)$ for $K$ times, with $K$ being the expansion ratio. For each latent feature, we inject perturbation over the latent feature $f$ with randomly initialized noise $z \sim \mathcal{U}(0, 1)$ and bias $b \sim \mathcal{N}(0, 1)$. Here, to prevent out-of-control imagination, we conduct residual multiplicative perturbation on the latent feature $f$ and enforce an $\varepsilon$-ball constraint on the perturbation as follows:

$$f' = \mathcal{P}_{f,\epsilon}((1 + z)f + b), \tag{1}$$

where $\mathcal{P}_{f,\epsilon}(\cdot)$ means to project the perturbed feature $f'$ to the $\varepsilon$-ball of the original latent feature, *i.e.,* $\|f' - f\|_\infty \leq \varepsilon$. Note that each latent feature has independent $z$ and $b$.

Following our explored criteria, GIF optimizes $z$ and $b$ over the latent feature space as follows:

$$z', b' \quad \leftarrow \quad \underset{z,b}{\arg\max} \quad \mathcal{S}_{con} + \mathcal{S}_{ent} + \mathcal{S}_{div}. \tag{2}$$

Here, $\mathcal{S}_{con}$, $\mathcal{S}_{ent}$ and $\mathcal{S}_{div}$ correspond to the class consistency, entropy difference and diversity, respectively. To compute these objectives, we resort to CLIP's zero-shot classification abilities. Specifically, we first use $f_{\text{CLIP-T}}$ to encode the class name $y$ of sample $x$ and take the embedding $w_y = f_{\text{CLIP-T}}(y)$ as the zero-shot classifier of class $y$. Each latent feature $f(x)$ can be classified according to its cosine similarity to $w_y$, *i.e.,* the affinity score of $x$ belonging to class $y$ is $s_y = \cos(f(x), w_y)$, which forms classification prediction vector $s = [s_1, \ldots, s_C]$ for the total $C$ classes of the target dataset. The prediction of the perturbed feature $s'$ can be obtained in the same way.

With the zero-shot prediction, we design the objective functions as follows: the *prediction consistency* $\mathcal{S}_{con}$ encourages the consistency between the predicted classification scores on $s$ and $s'$, so we define $\mathcal{S}_{con} = s'_i$, where $i = argmax(s)$ is the predicted class of the original latent feature. The criterion of *entropy maximization* $\mathcal{S}_{ent}$ seeks to improve the informativeness of the generated image, so we define $\mathcal{S}_{ent} = \text{Entropy}(s') - \text{Entropy}(s)$ to encourage the perturbed feature to have higher prediction entropy. To promote sample diversity, the *diversity* $\mathcal{S}_{div}$ is computed by the Kullback–Leibler (KL) divergence among all perturbed latent features of a seed sample: $\mathcal{S}_{div} = \text{KL}(f'; \bar{f})$, where $f'$ denotes the current perturbed latent feature and $\bar{f}$ indicates the mean over the $K$ perturbed latent features of this seed sample.

Table 1: Accuracy of ResNet-50 trained from scratch on small datasets and their expanded datasets by various methods. Here, CIFAR100-Subset is expanded by 5×, Pets is expanded by 30×, and all other datasets are expanded by 20×. Moreover, MAE and DALL-E2 are the baselines of directly using them to expand datasets without our guidance.

| Dataset | Caltech101 | Cars | Flowers | DTD | CIFAR100-S | Pets | Avg. |
|---|---|---|---|---|---|---|---|
| *Original* | $26.3_{\pm 1.0}$ | $19.8_{\pm 0.9}$ | $74.1_{\pm 0.2}$ | $23.1_{\pm 0.2}$ | $35.0_{\pm 1.7}$ | $6.8_{\pm 1.8}$ | 30.9 |
| KD with CLIP | $33.2_{\pm 1.1}$ | $18.9_{\pm 0.1}$ | $75.1_{\pm 0.8}$ | $25.6_{\pm 0.1}$ | $37.8_{\pm 3.1}$ | $11.1_{\pm 0.2}$ | 33.6 (+2.7) |
| *Expanded* | | | | | | | |
| Cutout (DeVries & Taylor, 2017) | $51.5_{\pm 0.7}$ | $25.8_{\pm 0.5}$ | $77.8_{\pm 0.1}$ | $24.2_{\pm 0.1}$ | $44.3_{\pm 1.6}$ | $38.7_{\pm 1.0}$ | 43.7 (+12.8) |
| GridMask (Chen et al., 2020) | $51.6_{\pm 0.6}$ | $28.4_{\pm 0.6}$ | $80.7_{\pm 0.8}$ | $25.3_{\pm 0.9}$ | $48.2_{\pm 0.7}$ | $37.6_{\pm 0.3}$ | 45.3 (+14.4) |
| RandAugment (Cubuk et al., 2020) | $57.8_{\pm 0.9}$ | $43.2_{\pm 2.1}$ | $83.8_{\pm 0.8}$ | $28.7_{\pm 1.2}$ | $46.7_{\pm 1.5}$ | $48.0_{\pm 0.4}$ | 51.4 (+20.5) |
| MAE (He et al., 2022) | $50.6_{\pm 0.8}$ | $25.9_{\pm 1.4}$ | $76.3_{\pm 0.5}$ | $27.6_{\pm 0.2}$ | $44.3_{\pm 1.7}$ | $39.9_{\pm 0.7}$ | 44.1 (+13.2) |
| DALL-E2 (Ramesh et al., 2022) | $61.3_{\pm 0.2}$ | $48.3_{\pm 0.3}$ | $84.1_{\pm 0.6}$ | $34.5_{\pm 0.4}$ | $52.1_{\pm 0.9}$ | $61.7_{\pm 0.9}$ | 57.0 (+26.1) |
| GIF-MAE (ours) | $58.4_{\pm 1.0}$ | $44.5_{\pm 0.3}$ | $84.4_{\pm 0.2}$ | $34.2_{\pm 0.2}$ | $52.7_{\pm 1.6}$ | $52.4_{\pm 0.5}$ | 54.4 (+23.5) |
| GIF-DALLE (ours) | $\mathbf{63.0}_{\pm 0.5}$ | $\mathbf{53.1}_{\pm 0.2}$ | $\mathbf{88.2}_{\pm 0.5}$ | $\mathbf{39.5}_{\pm 0.7}$ | $\mathbf{54.5}_{\pm 1.1}$ | $\mathbf{66.4}_{\pm 0.4}$ | **60.8 (+29.9)** |

Note that the above guided latent feature optimization is the key step for achieving guided imagination. After updating the noise $z'$ and bias $b'$ for each latent feature, GIF obtains a set of new latent features by Eq. (1), which are then used to create new samples through the decoder $G$. In this way, a small-scale dataset can be effectively expanded to a larger and more informative one.

Considering that DALL-E2 have proven to be powerful in generating images and MAE (He et al., 2022) is skilled at reconstructing images, we use them as prior models for imagination. We call the resulting methods GIF-DALLE and GIF-MAE, respectively.

GIF-DALLE follows exactly the aforementioned pipeline for guided imagination, while we slightly modify the pipeline of GIF-MAE because its encoder is not the CLIP image encoder. Specifically, GIF-MAE first generates a latent feature for the seed image based on its encoder, and conducts random *channel-level* noise perturbation following the way of Eq. (1). Based on the perturbed feature, GIF-MAE generates an intermediate image via its decoder, and applies CLIP to conduct zero-shot predictions for both the seed and the intermediate image to compute the guidance Eq. (2) for optimizing latent features. In this way, GIF-MAE can create content-consistent samples of diverse styles. More details of the two methods are provided in their pseudo codes in Appendix C.

## 5 EXPERIMENTS

### 5.1 EXPANSION ON NATURAL IMAGE DATASETS

**Settings.** We first evaluate the effectiveness of our proposed method on six small-scale natural image datasets, including object classification (Caltech-101 (Fei-Fei et al., 2004), CIFAR100-Subset (Krizhevsky et al., 2009)), fine-grained object classification (Standard Cars (Krause et al., 2013), Oxford 102 Flowers (Nilsback & Zisserman, 2008), Oxford-IIIT Pets (Parkhi et al., 2012)) and texture classification (DTD (Cimpoi et al., 2014)). Here, CIFAR100-subset is an artificial dataset for simulating small-scale datasets by randomly sampling 100 instances per class from the original CIFAR100 dataset. Their data statistics are given in Appendix D.

We implement GIF in PyTorch based on CLIP VIT-B/32, DALL-E2 and MAE VIT-L/16, which are pre-trained on large datasets and then fixed for dataset expansion. We use the official checkpoints of CLIP VIT-B/32 and MAE VIT-L/16, and use the DALL-E2 pre-trained on Laion-400M (Schuhmann et al., 2021). After expansion, we train ResNet-50 (He et al., 2016) from scratch for 100 epochs on the expanded datasets. More implementation details are provided in Appendix C.

As there is no algorithm devoted to dataset expansion, we take representative data augmentation methods as baselines, including RandAugment, Cutout, and GridMask (Chen et al., 2020). Besides, CLIP has demonstrated outstanding zero-shot ability thanks to its pre-training on extremely large-scale datasets, and some recent works explore distilling CLIP to facilitate model training. Hence, we also compare our method with knowledge distillation (KD) of CLIP on the original dataset.

**Comparisons with data augmentation.** As shown in Table 1, compared with the model trained on the original datasets, GIF-DALLE leads to 29.9% accuracy gain on average over six datasets, showing promising capabilities for expanding small datasets. Such improvement not only shows that dataset expansion is a promising direction for boosting DNNs on real small-data applications, but also verifies the effectiveness of our guided imagination in creating informative new samples.

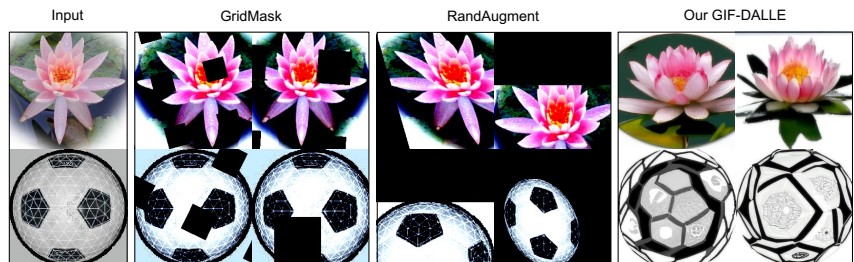

Figure 4: Accuracy of ResNet-50 trained from scratch on the expanded datasets with different expansion ratios. The performance is averaged over 3 runs. More results are reported in Appendix E.1.

Figure 5: Examples of the created samples for Caltech101 by data augmentation and GIF.

In addition, our expansion method is more *sample efficient* than data augmentations, in terms of the accuracy gain brought by each created sample. As shown in Figure 4, on Cars and DTD datasets, 5× expansion by GIF-DALLE even outperforms 20× expansion by various data augmentation methods, implying GIF-DALLE is 4× more sample efficient than them. This is because these augmentation methods cannot generate images with new and highly diversified content. In contrast, our GIF leverages strong prior models (*e.g.,* DALL-E2 and CLIP) trained from large datasets, and learns to perform imagination with the guidance from our discovered criteria. Hence, it can generate more diversified and informative images, leading to more significant performance gain.

**Visualization.** We further visualize the created samples. As shown in Figure 5, GridMask masks partial pixels of the inputs, while RandAugment randomly varies the images with a set of transformations. Both of them cannot create new content from the inputs, whereas our method can create images with new contents from the seed images, *e.g.,* water lilies with different postures and backgrounds. This further shows the superiority of our method. See more visualization in Appendix F.

**Comparisons to knowledge distillation by CLIP.** As shown in Table 1, although KD of CLIP indeed improves model performance on most datasets, it only leads to limited performance gain on those small-scale datasets. This reveals that simply distilling the knowledge of big models does not necessarily work well for the small-data regime. This also verifies the value of dataset expansion by creating informative new samples for model training on small datasets.

## 5.2 ANALYSIS

**Expanded datasets can boost various model architectures.** We further apply the expanded Cars dataset (5× expansion ratio) by GIF-DALLE to train ResNeXt-50 (Xie et al., 2017), WideResNet-50 (Zagoruyko & Komodakis, 2016) and MobileNet V2 (Sandler et al., 2018) from scratch. Table 2 shows that the expanded dataset brings consistent performance gain for all the architectures. This clearly demonstrates the effectiveness of the expanded dataset and the generalizability of the created samples by our method. Once expanded, the datasets are readily used for training other models.

Table 2: Performance of different model architectures trained on 5× expanded Cars by GIF.

| Dataset | ResNet-50 | ResNeXt-50 | WideResNet-50 | MobilteNet-v2 | Avg. |
|---|---|---|---|---|---|
| *Original* | $19.8_{\pm 0.9}$ | $18.4_{\pm 0.5}$ | $32.0_{\pm 0.8}$ | $26.2_{\pm 4.2}$ | 24.1 |
| *Expanded* | | | | | |
| RandAugment | $43.2_{\pm 2.1}$ | $29.6_{\pm 0.8}$ | $49.2_{\pm 0.2}$ | $39.7_{\pm 2.5}$ | 40.4 (+9.5) |
| GIF-DALLE | $\mathbf{53.1}_{\pm 0.2}$ | $\mathbf{43.7}_{\pm 0.2}$ | $\mathbf{60.0}_{\pm 0.6}$ | $\mathbf{47.8}_{\pm 0.6}$ | **51.2** (+27.1) |

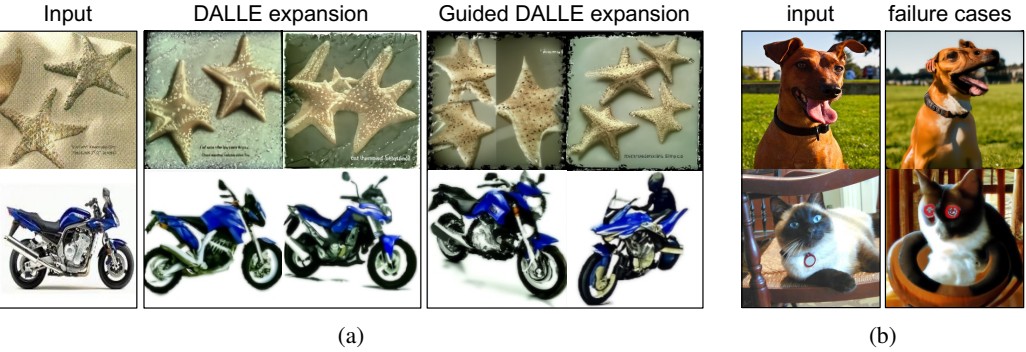

Figure 6: More examples of created samples by GIF-DALLE. (a) Visualization of DALLE expansion with and without our guidance. (b) Visualization of failure cases.

**Effectiveness of guided latent feature optimization.** Based on our guidance criteria, GIF optimizes the variation over the latent features for creating new samples. As shown in Table 1, our guided framework obtains consistent performance gain on all datasets compared to direct expansion with MAE or DALL-E2 respectively, which further demonstrates the effectiveness of our guidance criteria in optimizing informativeness and diversity of the created samples. More specifically, compared to direct DALLE expansion, as shown in Figure 6a, our GIF-DALLE with guidance can create starfish images with more diverse object numbers, and create motorbike images with more diverse angles of view and even a new driver. Moreover, compared to direct MAE expansion, GIF-MAE can generate informative samples with a more diverse style while maintaining image content, as shown in Figure 10 (cf. Appendix B). More ablation results of these criteria are given in Table 8 (cf. Appendix E.2).

**Pixel-wise vs. channel-wise noise.** GIF-MAE injects perturbation along the channel dimension instead of spatial dimension. Note that the generated image based on pixel-level noise variation is analogous to adding pixel-level noise to the original images. This may harm the integrity and smoothness of image content, leading the generated images to be noisy (cf. Figure 10(d)). In contrast, GIF decouples latent features into two dimensions (*i.e.,* token and channel), and particularly conducts channel-level perturbation. As shown in Figure 10(e), optimizing channel-level noise variation can generate more informative data, leading to more effective expansion (cf. Figure 7).

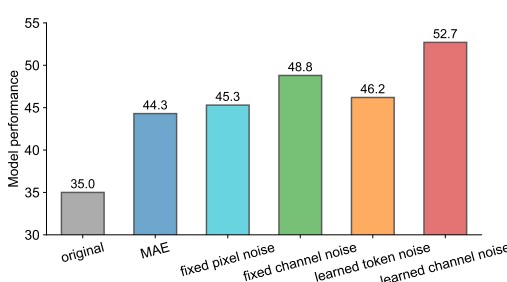

Figure 7: Effects of different kinds of noise on MAE for expanding CIFAR100-Subset by $5\times$.

**Failure cases.** We visualize some failure cases in Figure 6b. As we use pre-trained DALL-E2 without fine-tuning, the quality of some created samples may be limited due to domain shifts. For example, the face of the generated dog image in Figure 6b seems like a horse face. However, despite seeming less realistic, those samples are generated following our guidance criteria, so they can still maintain the class consistency and bring new information, thus being beneficial to model training.

## 5.3 APPLICATIONS TO MEDICAL IMAGE DATASETS

Medical image understanding is a severely data-lacking field. We thus explore to apply our method to expand three typical small-scale medical image datasets (Yang et al., 2021), including BreastMNIST (Al-Dhabyani et al., 2020), PathMNIST (Kather et al., 2019), and OrganSMNIST (Xu et al., 2019). These datasets cover a wide range of medical image modalities, including breast ultrasound (BreastMNIST), colon pathology (PathMNIST) and Abdominal CT (OrganSMNIST). See their detailed statistics in Appendix D. Considering that DALL-E2 was initially trained on natural images and suffers from domain shifts to medical domains (see discussion in Appendix E.3), we thus first fine-tune it on the target medical datasets, followed by dataset expansion. Similar to natural image

Table 3: Accuracy of ResNet-50 trained from scratch on small medical image datasets and their 5×-expanded datasets by various methods. All the performance is averaged over 3 runs.

| Dataset | PathMNIST | BreastMNIST | OrganSMNIST | Avg. |
|---|---|---|---|---|
| *Original* | 72.4 | 55.8 | 76.3 | 68.2 |
| KD with CLIP | 77.3 | 60.2 | 77.4 | 71.6 (+4.4) |
| *Expanded* | | | | |
| Cutout | 78.8 | 66.7 | 78.3 | 74.6 (+6.4) |
| GridMask | 78.4 | 66.8 | 78.9 | 74.7 (+6.5) |
| RandAugment | 79.2 | 68.7 | 79.6 | 75.8 (+7.6) |
| GIF-MAE | 82.0 | 73.3 | **80.6** | 78.6 (+10.4) |
| GIF-DALLE | **84.4** | **76.6** | 80.5 | **80.5** (+12.3) |

Table 4: Accuracy on the expanded OrganSMNIST with various expansion ratios.

| Expanded dataset | 5× | 10× | 20× |
|---|---|---|---|
| RandAugment | 79.6 | 80.1 | 80.5 |
| GIF-MAE | **80.6** | **81.1** | **81.2** |

Table 5: Comparison of model fine-tuning on OrganSMNIST with 20× expansion.

| | |
|---|---|
| *Original* | $76.3_{\pm 0.4}$ |
| Fine-tuning (ImageNet pre-trained) | $77.9_{\pm 0.6}$ |
| Fine-tuning (CLIP pre-trained) | $78.9_{\pm 0.1}$ |
| *Expanded* by GIF-MAE | $\mathbf{81.2}_{\pm 0.4}$ |

datasets, we train ResNet-50 from scratch based on the expanded datasets. The model performance is averaged over 3 runs in terms of macro accuracy.

**Comparisons with data augmentation.** As shown in Table 3, our GIF has a good ability to expand small-scale medical image datasets. Compared to the model trained on the original datasets, GIF-MAE brings 12.3% performance gains and GIF-MAE brings 10.4% performance gains on average over three medical image datasets. This further demonstrates the effectiveness and practicability of our method. Moreover, the result also reveals that, based on GIF, the prior MAE model pre-trained on large *natural image* datasets can also expand small *medical image* datasets well. In addition, our method also presents higher expansion efficiency than RandAugment on medical image datasets. As shown in Table 4, on OrganSMNIST, 10× expansion by GIF-MAE already outperforms 20× expansion by RandAugment. This further verifies the importance of generating more informative samples for medical dataset expansion, where we believe there is still huge room for improvement.

**Visualization.** We visualize the created medical samples in Figure 14 (cf. Appendix E.4). We find that RandAugment randomly varies the medical images based on a set of pre-defined transformations and may crop the lesion location of medical images, so it cannot guarantee the created samples to be informative and may even generate noisy samples. In contrast, our GIF-MAE can generate content-consistent images with diverse styles, and thus enrich medical images while retaining their lesion locations. Therefore, GIF-MAE expands medical datasets more effectively.

**Comparison with model fine-tuning.** The pre-training and fine-tuning scheme is widely used to transfer the representation ability learned from large datasets to small datasets. Here we also compare this scheme with our dataset expansion. As shown in Table 5, fine-tuning both the CLIP pre-trained or the ImageNet pre-trained ResNet-50 only leads to limited performance gain, which is worse than dataset expansion. That is, when the pre-trained datasets are highly different from the target dataset (*e.g.,* from natural images to medical images here), the pre-training and fine-tuning scheme does not significantly help performance, which was also verified by Raghu et al. (2019). In contrast, GIF can effectively exploit the knowledge of MAE and guide it to generate informative medical images, thus leading to better expansion effectiveness. This also reflects the importance of creating informative new samples for boosting real small-data medical applications.

## 6 CONCLUSIONS

This work has explored a new task, dataset expansion, towards resolving the challenging data scarcity issue for DNN training. Inspired by human learning with imagination, we presented a simple framework for dataset expansion via guided imagination. Promising empirical results on both small-scale natural and medical image datasets have demonstrated the effectiveness of the proposed method. Despite its encouraging results, there is significant room to improve our method. Specifically, as GIF does not fine-tune the pre-trained generative models on the target dataset, it may generate poor-quality images when there are huge domain shifts. Further fine-tuning the generative models can improve the quality of the expanded datasets, which is worth exploring in the future. Moreover, the expanded samples are still less informative than real samples. For example, we observed that ResNet-50 trained from scratch on our 5×-expanded CIFAR100-Subset (54.5±1.1) performs worse than on the original CIFAR100 dataset (71.0±0.6), indicating huge headroom for algorithmic dataset expansion to improve. We expect that this pioneering work can inspire future studies on dataset expansion, *e.g.,* how to leverage increasingly powerful generative models to achieve even better performance than human data collection.

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

# APPENDIX

## A    MORE RELATED STUDIES

**Image synthesis.** In the past decade, image synthesis has been widely studied following four main approaches: GANs (Isola et al., 2017; Esser et al., 2021), auto-regressive models (Kingma et al., 2019; Ramesh et al., 2021), diffusion models (Ho et al., 2020; Dhariwal & Nichol, 2021), and neural radiance fields (Mildenhall et al., 2020; Yu et al., 2021). For example, DALL-E2 (Ramesh et al., 2022) and Imagen (Saharia et al., 2022) trained on large-scale datasets have shown promising abilities to generate photo-realistic images, and thus can serve as generative models in our work for dataset expansion. Recently, CLIP (Radford et al., 2021), thanks to its text-image matching ability, has been used to guide image generation (Patashnik et al., 2021; Nichol et al., 2022; Kim et al., 2022; Wang et al., 2022), where CLIP is applied to match the generated image and a given text. In contrast, our work explores using CLIP to map latent features of generative models to the label space of the target small dataset, which enables them to conduct dataset expansion. In addition, model inversion (Xia et al., 2022) was also explored to generate images by inverting a trained classification network (Yin et al., 2020; Wang et al., 2021) or a GAN model (Zhu et al., 2020). Although our GIF framework only explores DALL-E2 and MAE (He et al., 2022) in this work, model inversion approaches can also be incorporated into our framework for dataset expansion, which is worth exploring in the future.

**More discussion on data augmentation.** Image data augmentation has been widely used to improve the generalization of DNNs during model training (Shorten & Khoshgoftaar, 2019; Yang et al., 2022). According to the technical characteristics, data augmentation for images can be divided into four categories: image manipulation, image erasing, image mix and auto augmentation. Specifically, image manipulation augments data via image transformations, including random flipping, random rotation, random scaling ratio, random cropping, random sharpening, random translation and so on (Yang et al., 2022). Image erasing augments data by randomly replacing pixel values of some image regions with constant values or random values, *e.g.,* Cutout (DeVries & Taylor, 2017), random erasing (Zhong et al., 2020), GridMask (Chen et al., 2020) and Fenchmask (Li et al., 2020). Image mix augments data by randomly mixing two or more images or sub-regions into one image, *e.g.,* Mixup (Zhang et al., 2021a), CutMix (Yun et al., 2019), and AugMix (Hendrycks et al., 2019). Auto augmentation seeks to augment images by automatically searching or randomly selecting augmentation operations from a set of random augmentations, including AutoAugment (Cubuk et al., 2019), Fast AutoAugment (Lim et al., 2019) and RandAugment (Cubuk et al., 2020).

Despite effectiveness in some applications, most of these augmentation methods impose pre-defined transformations to very each sample for enriching datasets, which only locally varies the pixel values of images and cannot generate images with highly diversified content. Moreover, as most methods are random augmentation operations, they cannot make sure the augmented samples are informative for model training and may even introduce noisy augmented samples. As a result, the brought new information can be limited to small datasets and the efficiency of dataset expansion is low. In comparison, the proposed GIF framework resorts to powerful generative models (*e.g.,* DALL-E2) trained on large-scale image datasets and guides them to conduct dataset expansion based on guided latent feature optimization designed by our discovered criteria (*i.e., zero-shot prediction consistency, entropy maximization* and *diversity promotion*). As a result, the created images are more informative and diversified than simple image augmentation, thus leading to more effective and efficient dataset expansion.

Note that, Xu et al. (2022) also explores MAE for image augmentation based on its reconstruction ability. Specifically, it augments images by first masking some sub-regions of images and then sending the masked images into MAE for image reconstruction. The reconstructed images with the recovered but slightly different sub-regions are then used for model training as augmented images. Like the above mentioned random augmentation methods, this approach (Xu et al., 2022) cannot generate images with new content and cannot ensure the reconstructed images to be informative and useful. In contrast, our GIF-MAE guides MAE to create informative new samples of diverse styles through our guided latent feature optimization strategy. Therefore, GIF-MAE is able to generate more useful synthetic samples for effective dataset expansion.

**Difference from dataset distillation.** Dataset distillation (Wang et al., 2018; Zhao et al., 2021; Zhao & Bilen, 2021; 2022), also called dataset condensation, aims to *condense a large dataset to a small/comparable set* of synthetic samples, which is expected to train models to have lossless performance compared to the original dataset. Such a task is in the opposite direction of our dataset expansion task, which aims to *expand a small dataset to a larger and more informative one* by automatically generating informative and diversified new samples.

**Difference from model transfer learning.** Based on existing large datasets (*e.g.,* ImageNet (Deng et al., 2009; Ridnik et al., 2021)), numerous studies have explored model transfer learning, such as model fine-tuning (Li et al., 2019; Gunel et al., 2021; Zhang et al., 2021b), knowledge distillation (Hinton et al., 2015; Gou et al., 2021), and domain adaptation (Ganin & Lempitsky, 2015; Tzeng et al., 2017). Despite effectiveness in some applications, these model transfer learning paradigms also suffer from key limitations. For example, Raghu et al. (2019) found that the pre-training and fine-tuning scheme does not significantly help model performance when the pre-trained datasets are very different from the target datasets (*e.g.,* from natural images to medical images). Moreover, model domain adaptation usually requires that the source dataset and the target dataset are paired with the same or highly overlapping label spaces, which is usually unsatisfiable in small-data application scenarios. In addition, Stanton et al. (2021) found that knowledge distillation does not necessarily work if the issue of model mismatch exists (Cho & Hariharan, 2019), *i.e.,* large discrepancy between the predictive distributions of the teacher model and the student model. The above limitations of model transfer learning also reflect the importance of effective dataset expansion: if a small dataset is effectively expanded, then various models can be directly trained on it. We note that some data-free knowledge distillation studies (Yin et al., 2020; Chawla et al., 2021) also synthesize images, but their goal is particularly to enable *knowledge distillation* in the setting without data. In contrast, our task is independent of model knowledge distillation. The expanded datasets are not method-dependent or model-dependent, and thus can train various model architectures to perform better than the original small ones.

# B MORE PRELIMINARY STUDIES

**Sample-wise expansion or sample-agnostic expansion**? When we design the selective expansion strategy in Section 3.2, another question appears in front of us: should we ensure that each sample is expanded by the same ratio? To determine this, we empirically compare RandAugment expansion with sample-wise selection and sample-agnostic selection according to two expansion criteria, *i.e., zero-shot prediction consistency* and *entropy maximization*. Figure 8 shows that sample-wise expansion performs much better than sample-agnostic expansion. To find out the reason for this phenomenon, we visualize how many times a sample is expanded by sample-agnostic expansion. As shown in Figure 9, the expansion numbers of different samples by sample-agnostic expansion present a long-tailed distribution, where many image samples are even not expanded. The main reason is that, due to the randomness of RandAugment and the differences among images, not all created samples are informative and it is easier for some samples to be augmented more effectively than others. Therefore, given a fixed expansion ratio, the sample-agnostic expansion strategy, as it ignores the differences of images, tends to select more expanded samples for those easy-to-augment images. This leads sample-agnostic expansion to waste some valuable original samples for expansion (*i.e.,* loss of information) and also incurs a class-imbalance problem, thus resulting in worse performance in Figure 8. In contrast, sample-wise expansion can fully take the advantage of all the samples in the target small dataset and thus is more effective than sample-agnostic expansion, which should be considered when designing dataset expansion approaches.

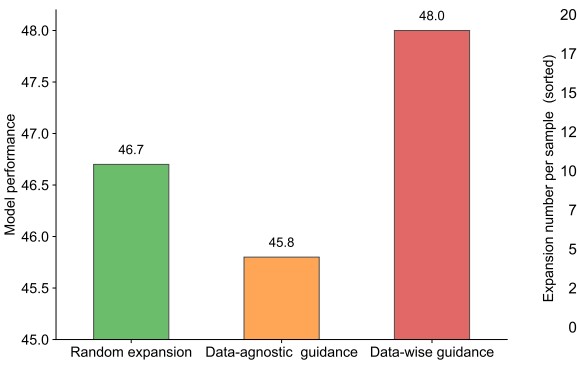 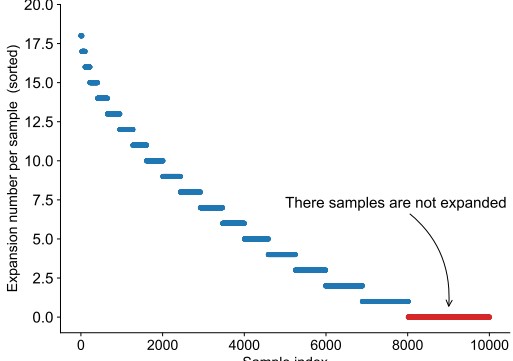

Figure 8: Comparison of model performance between samples-wise selection and sample-agnostic selection for RandAugment expansion on CIFAR100-Subset.

Figure 9: Statistics of the expansion numbers of different data in CIFAR100-Subset by sample-agnostic selective expansion with RandAugment, which presents a long-tailed distribution.

**Pixel-level noise or channel-level noise?** When we explore the MAE-based expansion strategy in preliminary studies, we first explore pixel-level noise to vary sample latent features, which, however, does not perform well. We dig the reason behind it by visualizing the reconstructed images. One illustrated example is given in Figure 10(d), from which we find that the generated image based on pixel-level noise variation is analogous to adding pixel-level noise to the original images. This may harm the integrity and smoothness of image content, leading the reconstructed images to be noisy and less informative. In comparison, as shown in Figure 10(b), the strong augmentation method (*i.e.,* RandAugment) mainly varies the style and geometric position of images, but slightly changes the content semantics of images, so it can better maintain the content consistency. This difference inspires us to factorize the influences on images into two dimensions: image styles and image contents. In light of Huang & Belongie (2017), we know that the channel-level latent feature encodes more subtle style information, whereas the token-level latent feature encodes more content information. We thus decouple the latent feature of MAE into two dimensions (*i.e.,* a token dimension and a channel dimension), and plot the latent feature distribution change between the generated image and the original image in these two dimensions.

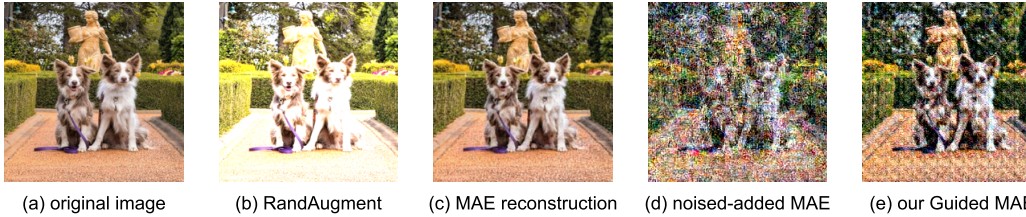

(a) original image     (b) RandAugment     (c) MAE reconstruction     (d) noised-added MAE     (e) our Guided MAE

Figure 10: An illustrated visualization of the generated images by (b) RandAugment, (c) MAE reconstruction, (d) random pixel-level variation over latent features, and (e) our guided MAE expansion. We find our guided MAE can generate content-consistent images of diverse styles.

We report the visualization in Figure 11. The added pixel-level noise changes the token-level latent feature distribution more significantly than RandAugment (cf. Figure 11(a)), but it only slightly changes the channel-level latent feature distribution (cf. Figure 11(b)). This implies that pixel-level noise mainly alters the content of images but slightly changes their styles, whereas RandAugment mainly influences the style of images while maintaining their content semantics. In light of this observation and the effectiveness of RandAugment, we are motivated to disentangle latent features into the two dimensions, and particularly conduct channel-level noise to optimize the latent features in our method. As shown in Figure 11, the newly explored channel-level noise variation varies the channel-level latent feature more significantly than the token-level latent feature, and thus can diversify the style of images while maintaining the integrity of image content. This innovation enables the explored MAE expansion strategy to generate more informative samples compared to pixel-level noise variation (cf. Figure 10(d) vs Figure 10(e)), leading to more effective dataset expansion, as shown in Figure 7 in the main paper.

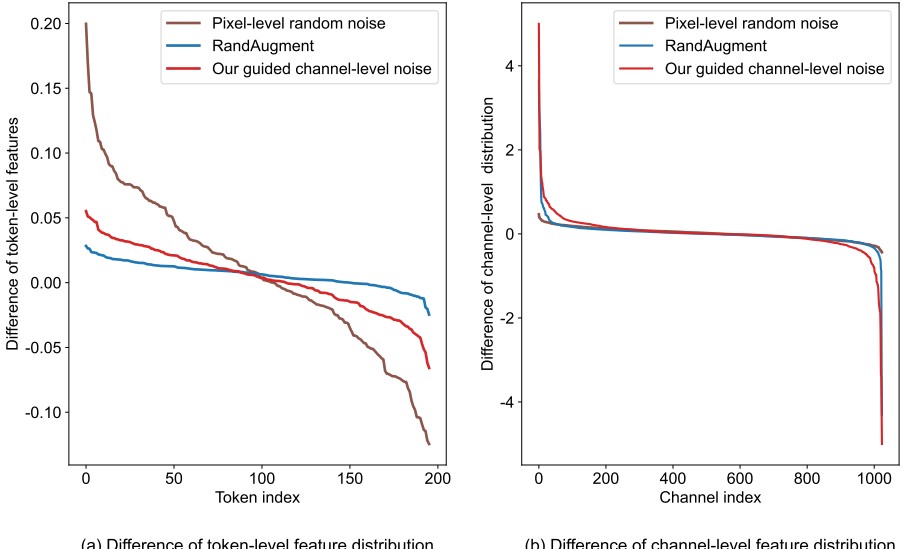

(a) Difference of token-level feature distribution     (b) Difference of channel-level feature distribution

Figure 11: Changes of the latent feature distributions along the token dimension and the channel dimension, between the latent feature of the generated image and that of the original image.

## C   MORE IMPLEMENTATION DETAILS

### C.1   IMPLEMENTATION DETAILS OF GIF-DALLE

Our GIF-DALLE applies DALL-E2 (Ramesh et al., 2022) as its prior generative model, which follows the pipeline described in Section 4. Its pseudo-code is provided in Algorithm 1. Specifically, DALL-E2 is built by using CLIP image/text encoders $f_{\text{CLIP-I}}$ and $f_{\text{CLIP-T}}$ as its image/text encoders and adopting a diffusion model $G$ as its decoder. Here, GIF-DALLE conducts guided imagination on the CLIP embedding space.

Here, we further clarify the implementation of the proposed guidance. Specifically, the *prediction consistency* $\mathcal{S}_{con}$ encourages the consistency between the predicted classification scores on $s$ and $s'$:

$$\mathcal{S}_{con} = s'_i, \quad \text{s.t.,} \quad i = \arg\max(s), \tag{3}$$

where $i = argmax(s)$ is the predicted class of the original latent feature. Such a loss helps to keep the prediction of the optimized feature to be the same as that of the original one. The criterion of *entropy maximization* $\mathcal{S}_{ent}$ seeks to improve the informativeness of the generated image as follows:

$$\mathcal{S}_{ent} = \text{Entropy}(s') - \text{Entropy}(s) = -s'\log(s') + s\log(s), \tag{4}$$

which encourages the perturbed feature to have higher information entropy regarding CLIP zero-shot predictions. To promote sample diversity, the *diversity* $\mathcal{S}_{div}$ is computed by the Kullback–Leibler (KL) divergence among all perturbed latent features of a seed sample as follows:

$$\mathcal{S}_{div} = \text{KL}(f'; \bar{f}), \tag{5}$$

where $f'$ denotes the current perturbed latent feature and $\bar{f}$ indicates the mean over the $K$ perturbed latent features of this seed sample. In implementing diversity promotion $\mathcal{S}_{div}$, we measure the dissimilarity of two feature vectors by applying the softmax function to the latent features, and then measuring the KL divergence between the resulting probability vectors.

In our experiment, the DALL-E2 model is pre-trained on Laion-400M (Schuhmann et al., 2021) and then fixed for dataset expansion. The resolution of the created images by GIF-DALLE is $64\times64$ for all datasets. Moreover, we set $\varepsilon = 0.1$ in the guided latent feature optimization.

---

**Algorithm 1:** GIF-DALLE Algorithm

**Input:** Original small dataset $\mathcal{D}_o$; CLIP image encoder $f_{\text{CLIP-I}}(\cdot)$; DALL-E2 diffusion decoder $G(\cdot)$; CLIP
   zero-shot classifier $w(\cdot)$; Expansion ratio $K$; Perturbation constraint $\varepsilon$.
**Initialize**: Synthetic data set $\mathcal{D}_s = \emptyset$;
**for** $x \in \mathcal{D}_o$ **do**
   $\mathcal{S}_{con} = \mathcal{S}_{ent} = 0$;
   $f = f_{\text{CLIP-I}}(x)$ ;                                    // latent feature
   $s = w(f)$ ;                                    // CLIP zero-shot prediction
   **for** $i=1,...,K$ **do**
      Initialize noise $z_i \sim \mathcal{U}(0,1)$ and bias $b_i \sim \mathcal{N}(0,1)$;
      $f'_i = \mathcal{P}_{f,\epsilon}((1+z_i)f + b_i)$ ;                   // noise perturbation
      $s' = w(f'_i)$;
      $\mathcal{S}_{con} += s'_j$,  w.r.t.  $j = \arg\max(s)$ ;          // prediction consistency score
      $\mathcal{S}_{ent} += \text{Entropy}(s') - \text{Entropy}(s)$ ;          // entropy difference score
   **end**
   $\bar{f} = mean(\{f'_i\}_{i=1}^K)$;
   $\mathcal{S}_{div} = sum(\{\text{KL}(f'_i; \bar{f})\}_{i=1}^K)$ ;                   // diversity score
   $\{z'_i, b'_i\}_{i=1}^K \leftarrow \arg\max_{z,b} \mathcal{S}_{con} + \mathcal{S}_{ent} + \mathcal{S}_{div}$;
   **for** $i=1,...,K$ **do**
      $f''_i = \mathcal{P}_{f,\epsilon}((1+z'_i)f + b'_i)$;
      $x''_i = G(f''_i)$;
      Add $x''_i \rightarrow \mathcal{D}_s$;
   **end**
**end**
**Output:** Expanded dataset $\mathcal{D}_o \cup \mathcal{D}_s$.

---

## C.2 IMPLEMENTATION DETAILS OF GIF-MAE

Our GIF-MAE applies the MAE-trained model (He et al., 2022) as its prior generative model based on its strong image reconstruction abilities. As its encoder is not the CLIP image encoder, we slightly modify the pipeline of GIF-MAE. As shown in Algorithm 2, GIF-MAE first generates a latent feature for the seed image based on its encoder, and conducts *channel-wise* noise perturbation. Here, the latent feature of MAE has two dimensions: spatial dimension and channel dimension. As discussed in our preliminary (cf. Appendix B), the channel-level latent feature encodes more subtle style information, whereas the token-level latent feature encodes more content information. Based on the finding in this preliminary study, we particularly conduct channel-level noise to optimize the latent features in our GIF-MAE method for maintaining the content semantics of images unchanged. Based on the perturbed feature, GIF-MAE generates an intermediate image via its decoder, and applies CLIP to conduct zero-shot prediction for both the seed and the intermediate image to compute the guidance $\mathcal{S}_{con}$, $\mathcal{S}_{ent}$ and $\mathcal{S}_{div}$. With these guidance, GIF-MAE optimizes the latent features for creating content-consistent samples of diverse styles. Here, GIF-MAE conducts guided imagination on its own latent space.

In our experiment, we implement GIF-MAE based on CLIP VIT-B/32 and MAE VIT-L/16, which are pre-trained on large datasets and then fixed for dataset expansion. Here, we use the official checkpoints of CLIP VIT-B/32 and MAE VIT-L/16. The resolution of the created images by GIF-MAE is $224 \times 224$ for all datasets. Moreover, we set $\varepsilon = 5$ for guided latent feature optimization in GIF-MAE.

---

**Algorithm 2:** GIF-MAE Algorithm

**Input:** Original small dataset $\mathcal{D}_o$; MAE image encoder $f(\cdot)$ and image decoder $G(\cdot)$; CLIP image
    encoder $f_{\text{CLIP-I}}(\cdot)$; CLIP zero-shot classifier $w(\cdot)$; Expansion ratio $K$; Perturbation constraint $\varepsilon$.
**Initialize**: Synthetic data set $\mathcal{D}_s = \emptyset$;
**for** $x \in \mathcal{D}_o$ **do**
  $\mathcal{S}_{con} = \mathcal{S}_{ent} = 0$;
  $f = f(x)$ ;                  // latent feature
  $s = w(f_{\text{CLIP-I}}(x))$ ;          // CLIP zero-shot prediction
  **for** $i=1,...,K$ **do**
    Initialize noise $z_i \sim \mathcal{U}(0,1)$ and bias $b_i \sim \mathcal{N}(0,1)$;
    $f_i' = \mathcal{P}_{f,\epsilon}((1+z_i)f + b_i)$ ;   // channel-level noise perturbation
    $x_i' = G(f_i')$ ;            // intermediate image generation
    $s' = w(f_{\text{CLIP-I}}(x_i'))$;
    $\mathcal{S}_{con} \mathrel{+}= s_j'$, w.r.t. $j = \arg\max(s)$ ;  // prediction consistency score
    $\mathcal{S}_{ent} \mathrel{+}= \text{Entropy}(s') - \text{Entropy}(s)$ ;  // entropy difference score
  **end**
  $\bar{f} = mean(\{f_i'\}_{i=1}^K)$;
  $\mathcal{S}_{div} = sum(\{\text{KL}(l_i'; \bar{f})\}_{f=1}^K)$ ;      // diversity score
  $\{z_i', b_i'\}_{i=1}^K \leftarrow \arg\max_{z,b} \mathcal{S}_{con} + \mathcal{S}_{ent} + \mathcal{S}_{div}$;
  **for** $i=1,...,K$ **do**
    $f_i'' = \mathcal{P}_{f,\epsilon}((1+z_i')f + b_i')$;
    $x_i'' = G(f_i'')$;
    Add $x_i'' \to \mathcal{D}_s$;
  **end**
**end**
**Output:** Expanded dataset $\mathcal{D}_o \cup \mathcal{D}_s$.

---

### C.3 MORE IMPLEMENTATION DETAILS OF MODEL TRAINING

We implement both GIF-DALLE and GIF-MAE in PyTorch. To fairly evaluate the expansion effectiveness of different methods, we use these methods to expand the original small datasets by the same ratios, followed by training models from scratch on the expanded dataset with the same number of epochs and same data pre-processing. In this way, the models are trained with the same number of update steps, so that all expansion methods are fairly compared.

The expansion ratio depends on the actual demand of real applications. In the experiment of Table 1, CIFAR100-Subset is expanded by 5×, Pets is expanded by 30×, and all other datasets are expanded by 20×. Moreover, in the experiment of Table 3, all medical image datasets are expanded by 5×. Moreover, all augmentation baselines expand datasets with the same expansion ratio for fair comparisons. After expansion, we train ResNet-50 (He et al., 2016) from scratch for 100 epochs based on the expanded datasets. During model training, we process images via random resize to 224×224 through bicubic sampling, random rotation and random flips. If not specified, we use the SGD optimizer with a momentum of 0.9. We set the initial learning rate (LR) to 0.01 with cosine LR decay, except the initial LR of CIFAR100-subset and OrganSMNIST is 0.1.

# D  DATASET STATISTICS

**The statistics of natural image datasets.** We evaluate our method on six small-scale natural image datasets, including Caltech-101 (Fei-Fei et al., 2004), CIFAR100-Subset (Krizhevsky et al., 2009), Standard Cars (Krause et al., 2013), Oxford 102 Flowers (Nilsback & Zisserman, 2008), Oxford-IIIT Pets (Parkhi et al., 2012) and DTD (Cimpoi et al., 2014). Here, CIFAR100-subset is an artificial dataset for simulating small-scale datasets by randomly sampling 100 instances per class from the original CIFAR100 dataset, and the total sample number is 10,000. These datasets cover a wide range of classification tasks, including coarse-grained object classification (*i.e.,* CIFAR100-subset and Caltech-101), fine-grained object classification (*i.e.,* Cars, Flowers and Pets) and texture classification (*i.e.,* DTD). The data statistics of these natural image datasets are given in Table 6. Note that the higher number of classes or the lower number of average samples per class a dataset has, the more challenging the dataset is.

Table 6: Statistics of small-scale natural image datasets.

| Datasets | Tasks | # Classes | # Samples | # Average samples per class |
|---|---|---|---|---|
| Caltech101 | Coarse-grained object classification | 102 | 3,060 | 30 |
| CIFAR100-Subset | Coarse-grained object classification | 100 | 10,000 | 100 |
| Standard Cars | Fine-grained object classification | 196 | 8,144 | 42 |
| Oxford 102 Flowers | Fine-grained object classification | 102 | 6,552 | 64 |
| Oxford-IIIT Pets | Fine-grained object classification | 37 | 3,842 | 104 |
| Describable Textures (DTD) | Texture classification | 47 | 3,760 | 80 |

**The statistics of medical image datasets.** To evaluate the effect of dataset expansion on medical images, we conduct experiments on three small-scale medical image datasets. These datasets cover a wide range of medical image modalities, including breast ultrasound (*i.e.,* BreastMNIST (Al-Dhabyani et al., 2020)), colon pathology (*i.e.,* PathMNIST (Kather et al., 2019)) and Abdominal CT (*i.e.,* OrganSMNIST (Xu et al., 2019)). We provide detailed statistics for these datasets in Table 7.

Table 7: Statistics of small-scale medical image datasets. To better simulate the scenario of small medical datasets, we use the validation sets of BreastMNIST and PathMNIST for experiments instead of training sets, whereas OrganSMNIST is based on its training set.

| Datasets | Data Modality | # Classes | # Samples | # Average samples per class |
|---|---|---|---|---|
| BreastMNIST (Al-Dhabyani et al., 2020; Yang et al., 2021) | Breast Ultrasound | 2 | 78 | 39 |
| PathMNIST (Kather et al., 2019; Yang et al., 2021) | Colon Pathology | 9 | 10,004 | 1,112 |
| OrganSMNIST (Xu et al., 2019; Yang et al., 2021) | Abdominal CT | 11 | 13,940 | 1,267 |

# E   MORE EXPERIMENTAL RESULTS AND DISCUSSION

## E.1   MORE COMPARISONS TO DATASET EXPANSION WITH AUGMENTATIONS

In Figure 4, we have demonstrated the expansion efficiency of our proposed GIF-DALLE over Cutout, GridMask and RandAugment on Cars and DTD and Pets datasets. Here, we further report the results on Caltech101, Flowers and CIFAR100-Subset datasets. As shown in Figure 12, 10× expansion by GIF-DALLE outperforms 20× expansion of these augmentation methods a lot, while 5× expansion by GIF-DALLE has already performed comparably to 20× expansion of these augmentation methods. This result further demonstrates the effectiveness and efficiency of our proposed method, and also reflects the importance of automatically creating informative synthetic samples for model training.

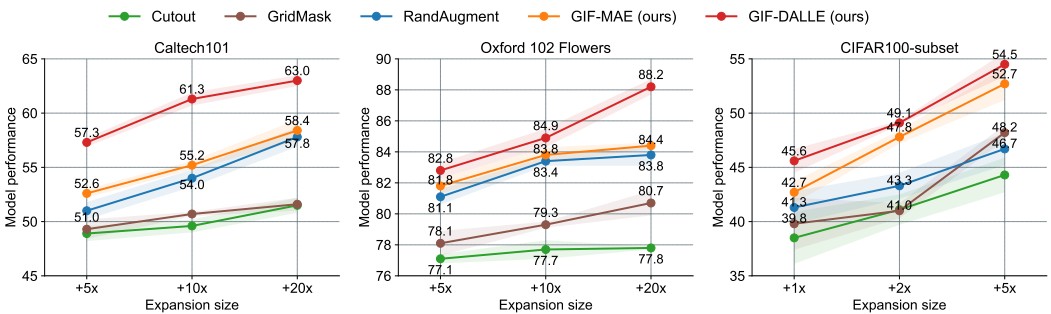

Figure 12: Accuracy of ResNet-50 trained from scratch on the expanded datasets with different expansion ratios based on Caltech101, Flowers and CIFAR100-Subset datasets.

## E.2   MORE ABLATION STUDIES ON THE GUIDANCE

GIF optimizes sample latent features for informative sample generation by maximizing the designed objective functions, *i.e.,* zero-shot prediction consistency $\mathcal{S}_{con}$, prediction entropy difference $\mathcal{S}_{ent}$, and diversity promotion $\mathcal{S}_{div}$. In Table 1, we have demonstrated their effectiveness in guiding DALL-E2 and MAE to generate informative samples for expansion. In this appendix, we further investigate their individual influence on GIF-DALLE for expanding CIFAR100-Subset by 5×.

As $\mathcal{S}_{con}$ and $\mathcal{S}_{ent}$ are closely coupled based on CLIP's zero-shot prediction for consistent-class information promotion, we regard them as a whole. As shown in Table 8, $\mathcal{S}_{con} + \mathcal{S}_{ent}$ is the foundation of effective expansion, since it makes sure that the created samples have correct labels and bring more information. Without it, only $\mathcal{S}_{div}$ cannot guarantee the created samples to be meaningful, although the sample diversity is improved, leading to even worse performance. In contrast, with $\mathcal{S}_{con} + \mathcal{S}_{ent}$, diversity promotion $\mathcal{S}_{div}$ can further bring more diverse information to boost data informativeness and thus achieve better performance (cf. Table 8). Note that entropy maximization $\mathcal{S}_{ent}$ and diversity promotion $\mathcal{S}_{div}$ play different roles. Entropy maximization promotes the informativeness of each generated image by increasing the prediction difficulty over the corresponding seed image, but this guidance cannot diversify different latent features obtained from the same image. By contrast, the guidance of diversity promotion encourages the diversity of various latent features of the same seed image, but it cannot increase the informativeness of generated samples regarding prediction difficulty. Therefore, using the two guidance together leads the expanded images to be more informative and diversified, thus bringing higher performance improvement.

Table 8: Ablation of guidance in GIF-DALLE for expanding CIFAR100-Subset by 5×.

| Method | $\mathcal{S}_{con}$ | $\mathcal{S}_{ent}$ | $\mathcal{S}_{div}$ | CIFAR100-Subset |
|---|---|---|---|---|
| | | | | $52.1_{\pm 0.9}$ |
| | ✓ | | | $52.8_{\pm 0.5}$ |
| | | ✓ | | $52.4_{\pm 0.5}$ |
| GIF-DALLE | | | ✓ | $51.8_{\pm 1.3}$ |
| | ✓ | ✓ | | $53.1_{\pm 0.3}$ |
| | ✓ | ✓ | ✓ | $54.5_{\pm 1.1}$ |

### E.3  ANALYSIS OF DALL-E2 ON MEDICAL IMAGES

DALL-E2 is trained on a large-scale dataset consisting of natural image and text pairs, and has shown powerful capabilities in natural image generation and variation. However, when we directly apply it to expand medical image datasets, we find the performance improvement is limited, compared to MAE as shown in Table 9. To pinpoint the reason, we visualize the generated images on OrganSMNIST in Figure 13 and find that it fails to generate photo-realistic medical images. The main reason is that DALL-E2 suffers from domain shifts between natural and medical images and cannot generate photo-realistic and informative medical samples based on its image variation abilities. In contrast, MAE is a reconstruction model and does not need to generate new content for the target images as DALL-E2, so it has much less negative impact by domain shifts. To address the issue, when applying DALL-E2 to medical domains, we first fine-tune it on target medical datasets, followed by dataset expansion. As shown in Table 9, based on the fine-tuned DALL-E2, GIF-DALLE can bring more significantly performance gains over GIF-MAE, and thus expands medical image datasets better.

Table 9: Accuracy of ResNet-50 trained on the $5\times$-expanded medical image datasets by GIF based on DALLE w/o and w/ fine-tuning. The performance is averaged over 3 runs.

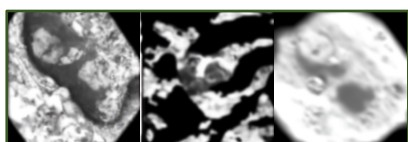

Figure 13: Visualization of the synthetic medical images by DALL-E2 via its variation abilities on the OrganSMNIST dataset.

| Dataset | PathMNIST | BreastMNIST | OrganSMNIST |
|---|---|---|---|
| *Original* | $72.4_{\pm 0.7}$ | $55.8_{\pm 1.3}$ | $76.3_{\pm 0.4}$ |
| GIF-MAE | $82.0_{\pm 0.7}$ | $73.3_{\pm 1.3}$ | $\mathbf{80.6}_{\pm 0.5}$ |
| GIF-DALLE (w/o tuning) | $78.4_{\pm 1.0}$ | $59.3_{\pm 2.5}$ | $76.4_{\pm 0.3}$ |
| GIF-DALLE (w/ tuning) | $\mathbf{84.4}_{\pm 0.3}$ | $\mathbf{76.6}_{\pm 1.4}$ | $80.5_{\pm 0.2}$ |

### E.4  VISUALIZATION OF CREATED MEDICAL IMAGES

We visualize the created medical samples by different methods. As shown in Figure 14, RandAugment randomly varies the medical images based on a set of pre-defined transformations. However, due to its randomness, RandAugment may crop the lesion location of medical images and cannot guarantee the created samples to be informative, even leading to noise samples. In contrast, our GIF-MAE can generate content-consistent images with diverse styles, so it can enrich the medical images while maintaining their lesion location unchanged. Therefore, GIF-MAE is able to expand medical image datasets better than RandAugment, leading to higher model performance improvement (cf. Table 3 in the main paper).

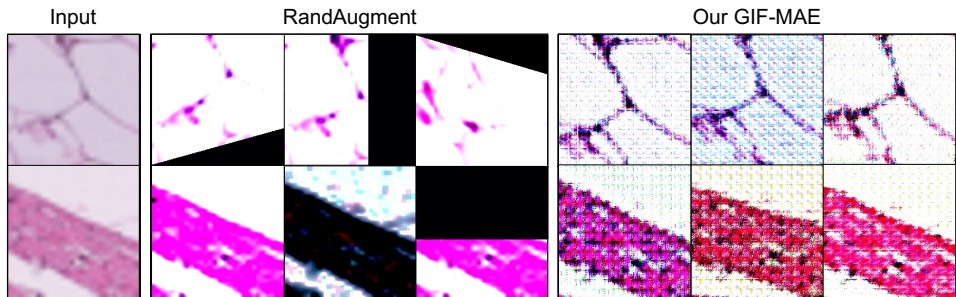

Figure 14: Examples of the created samples for PathMNIST by RandAugment and GIF-MAE.

### E.5 DISCUSSION ON THE USE OF ZERO-SHOT CLIP

In GIF, we exploit the zero-shot discriminability of the pre-trained CLIP to guide dataset expansion. It is interesting to know whether fine-tuning CLIP on the target small dataset can bring further improvement. To determine this, we further compare the results of GIF-MAE with fine-tuned CLIP and that with zero-shot CLIP based on OrganSMNIST. To be specific, we add an additional linear classifier on the top of the CLIP image encoder and fine-tune the CLIP model on OrganSMNIST, where the accuracy of the fine-tuned CLIP is 78.9. As shown in Table 10, GIF-MAE with fine-tuned CLIP performs only comparably to that with zero-shot CLIP. This reflects that the CLIP text classifier is enough to provide sound guidance based on its text-image matching abilities, whereas simply fine-tuning CLIP does not work better on small-scale medical datasets. Even so, we expect to further improve CLIP with more effective adaptation and prompt engineering in the future, which may better unleash the power of CLIP to guide dataset expansion.

Table 10: Comparison between the model performance by GIF-MAE expansion with zero-shot CLIP and fine-tuned CLIP based on the OrganSMNIST medical image dataset. Here, the expansion ratios are $\{5\times, 10\times, 20\times\}$. The performance is averaged over 3 runs.

| Dataset | OrganSMNIST | | |
|---|---|---|---|
| *Original* | $76.3_{\pm 0.4}$ | | |
| *Expanded* | $5\times$ | $10\times$ | $20\times$ |
| with fine-tuned CLIP | $80.1_{\pm 0.3}$ | $80.6_{\pm 0.2}$ | $81.0_{\pm 0.5}$ |
| with zero-shot CLIP (ours) | $\mathbf{80.6}_{\pm 0.5}$ | $\mathbf{81.1}_{\pm 0.5}$ | $\mathbf{81.2}_{\pm 0.4}$ |

### E.6 WHY NOT DIRECTLY TRANSFER CLIP MODELS TO TARGET DATASETS?

In our proposed GIF framework, we resort to the pre-trained CLIP to guide dataset expansion. One may wonder why not to directly pre-train and fine-tune CLIP models to the target datasets, which has shown effectiveness in many natural image datasets. Before discussing this, we would like to highlight that we explore dataset expansion to handle real small-data scenarios, where only a small-size dataset is available without any large-scale external dataset of similar image nature. Therefore, pre-training models with CLIP or other self-supervised methods on large-scale target datasets is inapplicable. In our proposed method, we resort to publicly available CLIP models for dataset expansion. Compared to directly transferring CLIP models, our dataset expansion is a necessarily new paradigm for the following two key reasons.

First, our GIF method has better applicability to the scenarios of different image domains. Although CLIP has strong transfer performance on some natural image datasets, its transfer performance on other domains like medical image datasets is limited. Here, we report the fine-tuning performance of CLIP-trained ResNet-50 model on three medical datasets. As shown in Table 11, transferring the CLIP model only leads to limited performance gains, which are significantly worse than our dataset expansion. The reason is that, when the pre-trained datasets are highly different from the target dataset, the pre-training and fine-tuning scheme does not significantly help performance (Raghu et al., 2019). In contrast, our GIF framework can generate images of similar nature as the target dataset for expansion, and thus GIF is more beneficial to real applications in various image domains.

Table 11: Comparison between our methods and directly fine-tuning CLIP models on three medical image datasets. The performance is averaged over 3 runs.

| Dataset | PathMNIST | BreastMNIST | OrganSMNIST |
|---|---|---|---|
| *Original* dataset | $72.4_{\pm 0.7}$ | $55.8_{\pm 1.3}$ | $76.3_{\pm 0.4}$ |
| Fine-tuned CLIP | $78.4_{\pm 0.9}$ | $67.2_{\pm 2.4}$ | $78.9_{\pm 0.1}$ |
| GIF-MAE | $82.0_{\pm 0.7}$ | $73.3_{\pm 1.3}$ | $\mathbf{80.6}_{\pm 0.5}$ |
| GIF-DALLE | $\mathbf{84.4}_{\pm 0.3}$ | $\mathbf{76.6}_{\pm 1.4}$ | $80.5_{\pm 0.2}$ |

Our dataset expansion can provide expanded datasets ready for training various network architectures. In some real application scenarios like mobile terminals, the supportable model size is very limited due to the constraints of device hardware. However, the publicly available checkpoints provided by CLIP are only ResNet50, ViT-B/32 or even larger models, which may not be allowed to use in those scenarios. By contrast, the expanded dataset by our method can be directly used to train various model architectures (cf. Table 2), and thus is more applicable to the application scenarios with hardware constraints. Also, once the datasets are expanded, they can be released to the public to facilitate future studies.

Please note that the goal of this paper is to show the huge potential of dataset expansion as a promising future direction instead of completely resolving it. We expect that the performance of dataset expansion can be further improved in future research.

### E.7 EFFECTIVENESS ON LARGER-SCALE DATASETS

In previous experiments, we have demonstrated the effectiveness of our proposed method on small-scale natural and medical datasets. One may be interested in whether our method can be applied to larger-scale datasets. Although expanding larger-scale datasets is not the goal of this paper, we also explore our method to expand the full CIFAR100 by $5\times$ for model training from scratch. As shown in Table 12, our GIF-DALLE leads to an 8.7% accuracy gain compared to direct training on the original CIFAR100 dataset. Such encouraging results verify the effectiveness of our methods on larger-scale datasets, and we expect to apply our methods to even larger datasets like ImageNet and other tasks in the future.

Table 12: Effectiveness of GIF-DALLE for expanding CIFAR100 by $5\times$.

| Dataset | CIFAR100 |
| --- | --- |
| *Original* | $70.9_{\pm0.6}$ |
| *Expanded* | |
| GIF-MAE | $77.0_{\pm0.3}$ |
| GIF-DALLE | $\mathbf{79.6}_{\pm0.3}$ |

### E.8 COMPARISON TO INFINITE DATA AUGMENTATION

In previous experiments, we have demonstrated that our proposed GIF framework is more effective and efficient than existing data augmentation methods in expanding small datasets. Despite this, one may also wonder how the explored dataset expansion would perform compared to training with infinite data augmentation. Therefore, in this appendix, we further evaluate the performance of infinite data augmentation on CIFAR100-subset. Specifically, based on RandAugment, we train ResNet-50 using infinite online augmentation for varying numbers of epochs from 100 to 700. As shown in Table 13, using RandAugment to train models for more epochs leads to better performance, but gradually converges (around 51% accuracy at 500 epochs) and keeps fluctuating afterward. By contrast, our GIF-DALLE can achieve better performance when only training 100 epochs, which further demonstrates the effectiveness of our method in generating informative synthetic data for model training.

Table 13: Comparison between GIF-DALLE and infinite data augmentation on CIFAR100-subset.

| Methods | Epochs | Accuracy |
| --- | --- | --- |
| *Original* | | |
| Standard training | 100 | $35.0_{\pm1.7}$ |
| Training with RandAugment | 100 | $39.6_{\pm2.5}$ |
| Training with RandAugment | 200 | $46.9_{\pm0.9}$ |
| Training with RandAugment | 300 | $48.1_{\pm0.6}$ |
| Training with RandAugment | 400 | $49.6_{\pm0.4}$ |
| Training with RandAugment | 500 | $51.3_{\pm0.3}$ |
| Training with RandAugment | 600 | $51.1_{\pm0.3}$ |
| Training with RandAugment | 700 | $50.6_{\pm1.1}$ |
| *Expanded* | | |
| *$5\times$-expanded* by GIF-DALLE | 100 | $\mathbf{54.5}_{\pm1.1}$ |

### E.9 DISCUSSION OF PICKING RELATED SAMPLES FROM LARGER DATASETS

Picking and labeling data from larger image datasets with CLIP is an interesting idea for dataset expansion. However, such a solution is limited in real applications, since a large-scale related dataset may be unavailable in many image domains. Moreover, selecting data from different image domains (*e.g.,* from natural images to medical images) is unhelpful for dataset expansion.

Despite the above limitations in real applications, we also evaluate this idea on CIFAR100-subset and investigate whether it helps dataset expansion when there is a larger dataset of the same image nature, *e.g.,* ImageNet. Here, we use CLIP to select and annotate related images from ImageNet to expand CIFAR100-subset. Specifically, we scan over all ImageNet images and use CLIP to predict them to the class of CIFAR100-subset. We select the samples with the highest prediction probability higher than 0.1 and expand each class by 5×. As shown in Table 14, the idea of picking related images from ImageNet makes sense, but performs worse than our proposed method. This result further demonstrates the effectiveness and superiority of our method. In addition, how to better transfer large-scale datasets to expand small datasets is an interesting open question, and we expect to explore it in the future.

Table 14: Comparison between GIF and picking related data from ImageNet for expanding CIFAR100-subset by 5×.

| CIFAR100-Subset | Accuracy |
|---|---|
| *Original dataset* | $35.0_{\pm1.7}$ |
| *Expanded dataset* | |
| *5×-expanded* by picking data from ImageNet with CLIP | $50.9_{\pm1.1}$ |
| *5×-expanded* by GIF-DALLE | $\mathbf{54.5}_{\pm1.1}$ |

### E.10 DISCUSSION OF TRAINING MODELS WITH ONLY EXPANDED IMAGES

It is interesting to know how the model would perform when trained with only the expanded images by our method. More specifically, we use only the expanded images by GIF-DALLE on CIFAR100-subset to train ResNet-50 from scratch, and compare it to the model trained on real images of CIFAR100-subset. As shown in Table 15, the model trained with our synthetic images performs comparably to the model trained with real images. This result further verifies the effectiveness of our explored dataset expansion method. Moreover, the model trained with the full expanded dataset performs much better than that trained with only the original dataset or with only the generated images. That is, the generated images are not a simple repetition of the original dataset, but bring new information to the expanded dataset for model training. This further shows that using synthetic images for model training is a promising direction. We expect that our work on dataset expansion can inspire more studies to explore this direction in the future.

Table 15: Performance of the model trained with only the expanded images on the 5×-expanded dataset of CIFAR100-subset by GIF-DALLE.

| CIFAR100-Subset | Accuracy |
|---|---|
| *Training with real images in original dataset* | $35.0_{\pm1.7}$ |
| *Training with 5×-expanded dataset* by GIF-DALLE | $54.5_{\pm1.1}$ |
| *Training with only the expanded images* by GIF-DALLE | $35.2_{\pm1.3}$ |

### E.11 Comparison to CutMix for dataset expansion

In this appendix, we further apply a more advanced augmentation method, *i.e.,* CutMix (Yun et al., 2019), to expand CIFAR100-subset by 5 times and use the expanded dataset to train the model from scratch. The results in Table 16 further demonstrate the superiority of our method over augmentation-based expansion methods.

Table 16: Comparison between GIF-DALLE and CutMix (Yun et al., 2019) for expanding CIFAR100-subset by $5\times$.

| CIFAR100-Subset | Accuracy |
|---|---|
| *Original dataset* | $35.0_{\pm 1.7}$ |
| *Expanded dataset* | |
| $5\times$-*expanded* by CutMix | $50.7_{\pm 0.2}$ |
| $5\times$-*expanded* by GIF-DALLE | $\mathbf{54.5}_{\pm 1.1}$ |

# F    MORE VISUALIZATION RESULTS

This appendix provides more visualized results for the created samples by our methods on various natural image datasets. Specifically, we report the synthetic images by GIF-DALLE on Caltech101 in Figure 15 and those by GIF-MAE in Figure 16. The visualized results show that our GIF-DALLE can create semantic-consistent yet content-diversified images well, while GIF-MAE can generate content-consistent yet highly style-diversified images. In addition, the visualization of GIF-DALLE on other natural image datasets are shown in Figures 17-21.

## F.1    VISUALIZATION OF THE EXPANDED IMAGES ON CALTECH101

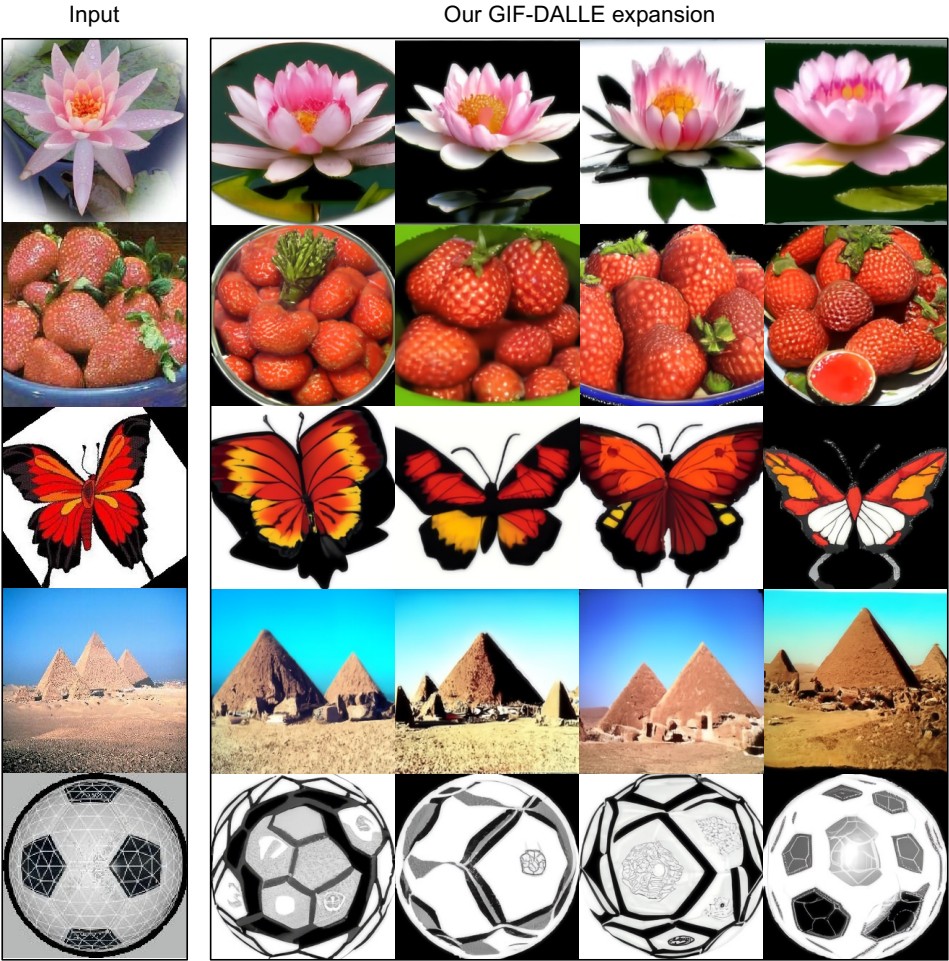

Figure 15: Visualization of the created samples on Caltech101 by GIF-DALLE.

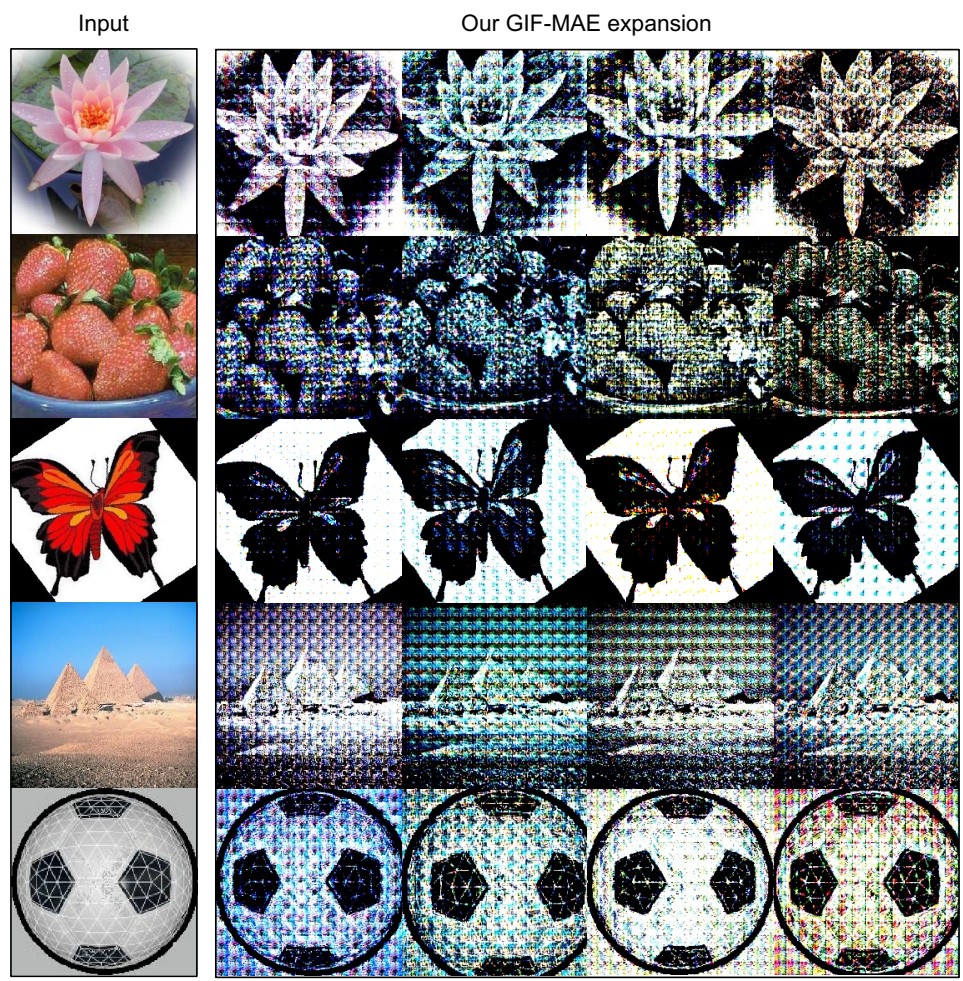

Figure 16: Visualization of the created samples on Caltech101 by GIF-MAE.

## F.2 VISUALIZATION OF THE EXPANDED IMAGES ON CARS

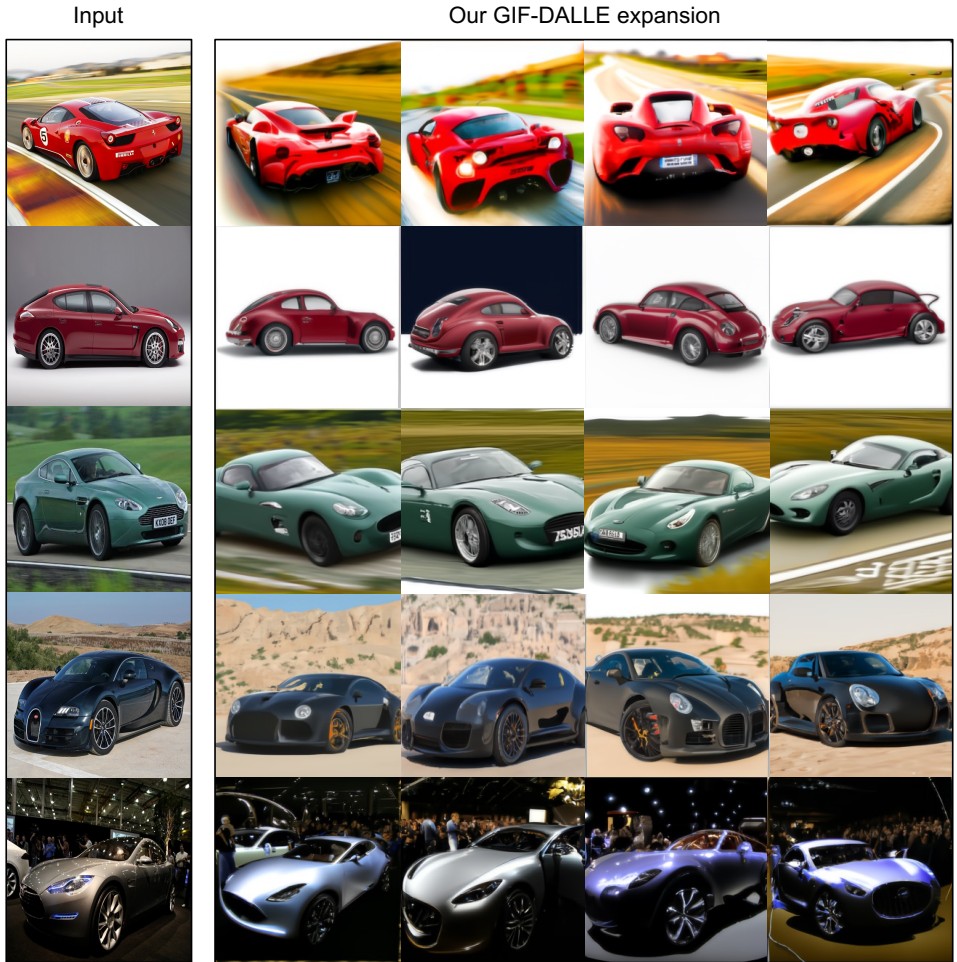

Figure 17: More visualization of the synthetic samples on Cars by GIF-DALLE.

## F.3 Visualization of the expanded images on Flowers

Input                           Our GIF-DALLE expansion

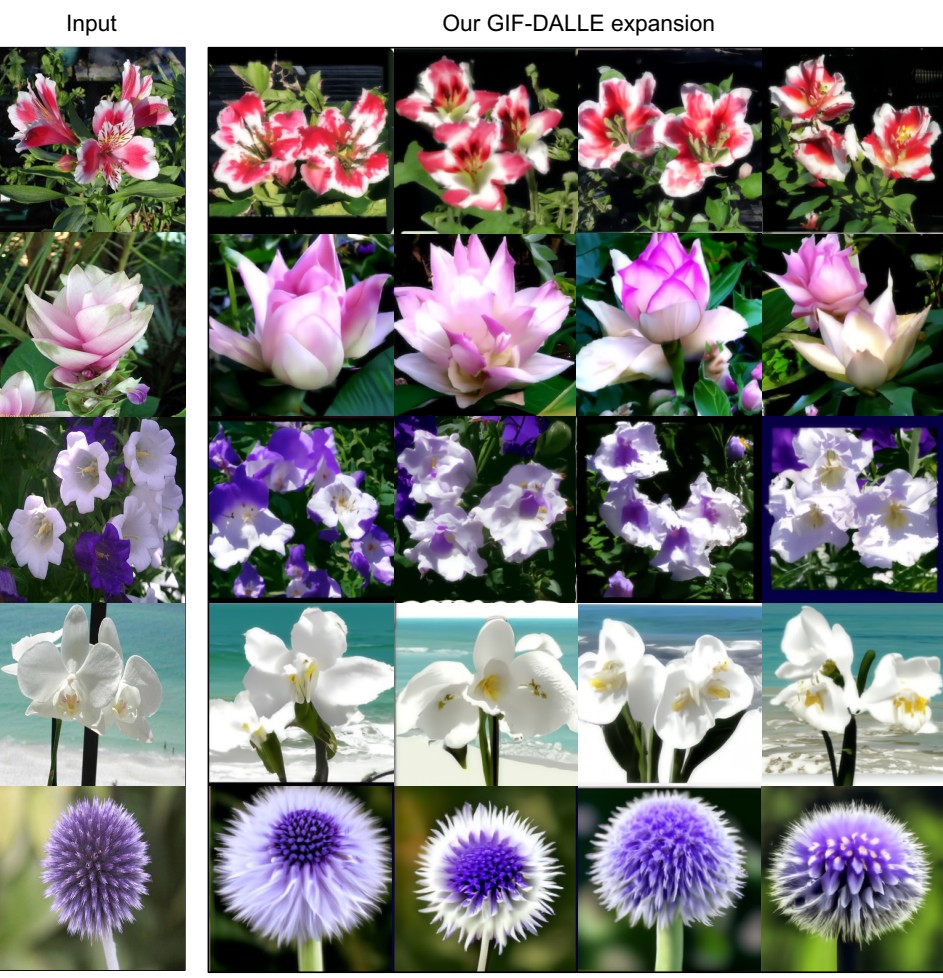

Figure 18: More visualization of the synthetic samples on Flowers by GIF-DALLE.

### F.4 VISUALIZATION OF THE EXPANDED IMAGES ON PETS

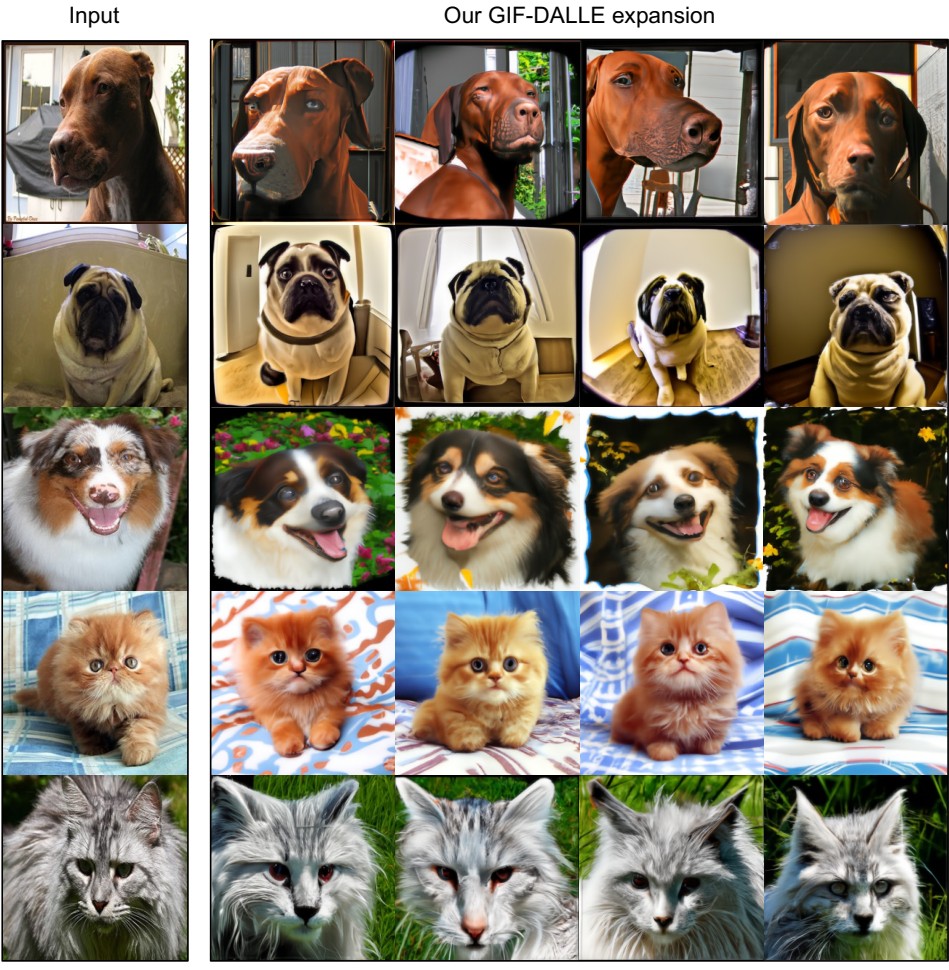

Figure 19: More visualization of the synthetic samples on Pets by GIF-DALLE.

## F.5 VISUALIZATION OF THE EXPANDED IMAGES ON CIFAR100-SUBSET

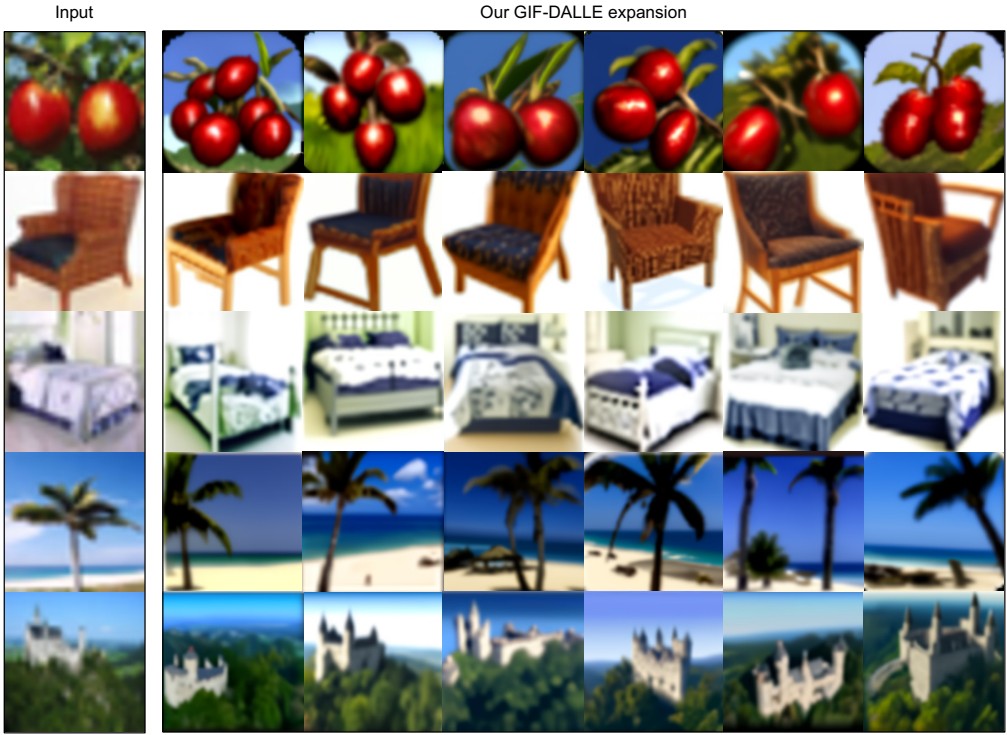

Figure 20: More visualization of the synthetic samples on CIFAR100-Subset by GIF-DALLE. Note that the resolution of CIFAR100 images is small (*i.e.,* 32×32), so their visualization is a little unclear.

## F.6 VISUALIZATION OF THE EXPANDED IMAGES ON DTD

Input                    Our GIF-DALLE expansion

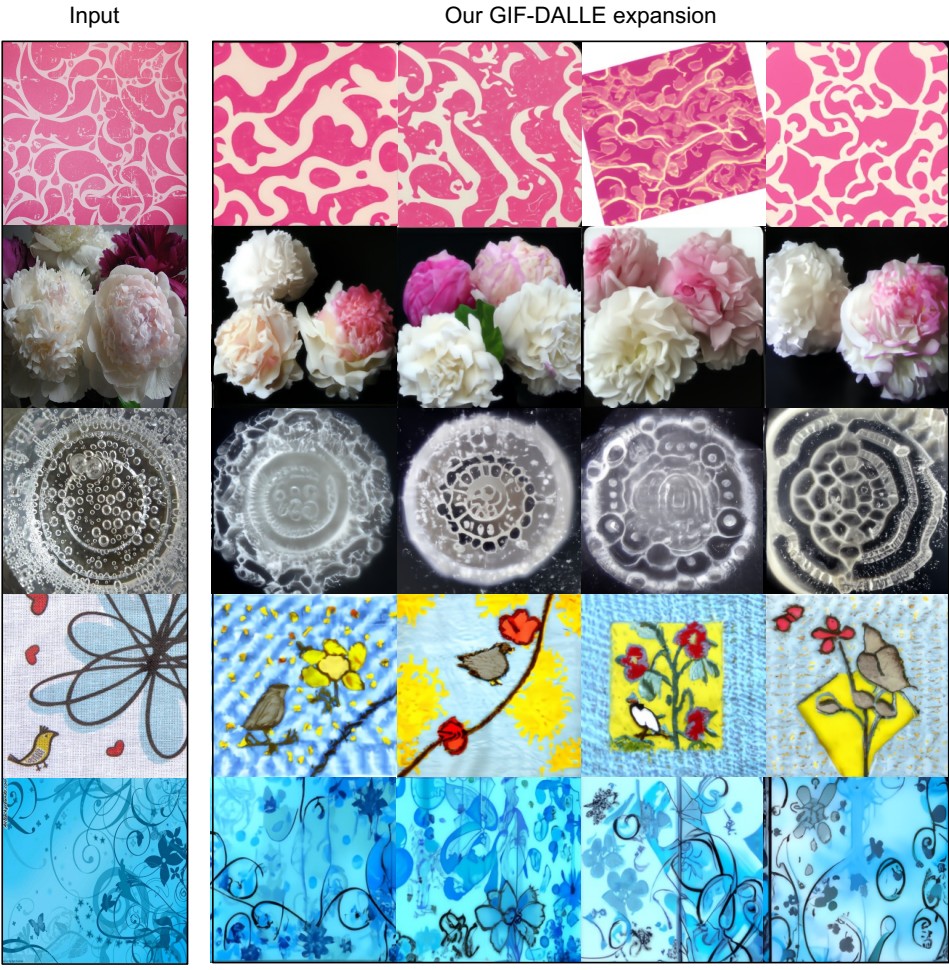

Figure 21: More visualization of the synthetic samples on DTD by GIF-DALLE.

## G  EXPLORATORY THEORETICAL ANALYSIS

In this work, we explore the empirical possibility of using recent generative models (*e.g.,* DALL-E2) for small dataset expansion, and we have empirically demonstrated the effectiveness of our proposed GIF method in previous experiments. Although theoretical analysis is not the focus of this paper, it is also interesting to theoretically analyze the proposed dataset expansion. However, rigid theoretical analysis for dataset expansion is complex and non-trivial, so here we provide an exploratory idea to analyze the benefits of our dataset expansion to generalization performance.

Inspired by Sener & Savarese (2017), we resort to the concept of $\delta$ cover to analyze how the diversity of data influences the generalization error bound. Specifically, "a dataset $s$ is a $\delta$ cover of a dataset $\hat{s}$" means a set of balls with radius $\delta$ centered at each sample of the dataset $s$ can cover the entire dataset $\hat{s}$. In our work, we only consider the small target dataset and its true data distribution, so we follow the assumptions of the work (Sener & Savarese, 2017) and extend its Theorem 1 to the version of the generalization error bound.

**Corollary 1** *Let $A$ be a learning algorithm and $C$ be a constant. Given a training set $\mathcal{D} = \{x_i, y_i\}_{i \in [n]}$ with $n$ \*i.i.d.\* samples drawn from the true data distribution $\mathcal{P}_{\mathcal{Z}}$. If the training set $\mathcal{D}$ is $\delta$ cover of the true distribution $\mathcal{P}_{\mathcal{Z}}$, the hypothesis function is $\lambda^{\eta}$-Lipschitz continuous, the loss function $\ell(x, y)$ is $\lambda^{\ell}$-Lipschitz continuous for all $y$ and bounded by $L$, and $\ell(x_i, y_i; A) = 0$ for $\forall i \in [n]$, with the probability at least $1 - \gamma$, the generalization error bound satisfies:*

$$|\mathbb{E}_{x,y \sim \mathcal{P}_{\mathcal{Z}}}[\ell(x, y; A)] - \frac{1}{n} \sum_{i \in [n]} \ell(x_i, y_i; A)| \leq \delta(\lambda^{\ell} + \lambda^{\eta} L C). \tag{6}$$

This corollary shows that the generalization error is bounded by the covering radius $\delta$. In real small-data applications, the data limitation issue leads $\delta$ to be very large and thus severely affects the generalization performance of the trained model. More critically, simply increasing the data number (*e.g.,* via data repeating) does not help the generalization since it does not decrease the covering radius $\delta$. Rather than simply increasing the number of samples, our proposed GIF framework guides recent generative models (*e.g.,* DALL-E2) to synthesize informative and diversified new images for expanding the original small dataset. Therefore, the expanded dataset would have higher diversity of data, which helps to decrease the covering radius $\delta$ and thus improves the trained model's generalization performance.

