# OpenReview forum: "Expanding Datasets With Guided Imagination"
_ICLR.cc/2023/Conference — Submitted to ICLR 2023_

### Official Review · Reviewer_sxmr · 2022-10-17

**Confidence:** 5
**Correctness:** 3
**Technical Novelty And Significance:** 3
**Empirical Novelty And Significance:** 1
**Recommendation:** 3

**Clarity, Quality, Novelty And Reproducibility:**

In terms of clarity and quality, this paper is well-written, well-organized, and easy to follow. I really enjoyed reading this paper.

The idea of augmentation by generation is not very new. However, I think the method contribution of this paper is fair. I think this paper discusses related previous work enough.

However, in terms of empirical novelty, this paper is neither significant nor novel. Even CLIP zero-shot outperforms the proposed method although the method heavily relies on the CLIP zero-shot prediction itself. Furthermore, if we allow access to training data, CLIP linear probe outperforms the proposed method with a significantly large gap.

I am a little bit concerned about reproducibility when the big models are not accessible. For example, as far as the reviewer understood, the Dall-E2 and MAE decoders are trained by the authors using LAION-400M. Training with LAION-400M is not always available by many researchers. Even if one can use the dataset (and enough GPUs), the reproducibility of large-scale training itself is often challenging. Except for this big model part, I think this paper is fairly reproducible with the attached pseudo code (C.1).

**Strength And Weaknesses:**

## Strength

- This paper leverages the knowledge from the models trained with large-scale datasets by generating more data samples
- Compared to previous data augmentation methods, the generated images look more "realistic" and seem helpful
- This paper provides enough analysis to show the effectiveness of the proposed method (e.g., various architectures, where to optimize, pixel-wise vs. channel-wise noise, ...)
- The paper is well-written and easy to follow.

## Weakness

### Comparison with CLIP zero-shot / linear-probe results

I think it is the most critical weakness of this paper. Note that the proposed method heavily relies on CLIP ViT-B/32 zero-shot performance and Dall-E2 decoder where both models are trained with 400M image-text pairs (CLIP is trained with private samples and Dall-E2 decoder is trained with LAION-400M, respectively). I checked the zero-shot performance of CLIP ResNet-50 and ViT-B/32 models and found that the proposed method is much worse than CLIP zero-shot performances (from the original CLIP paper [R1] Table 17):

| Methods                         | Caltech | Cars | Flowers | DTD | CIFAR100-S | Pets | Avg. |
|-----------------------------|-----------|------|----------|-------|---------------|-------|------|
| GIF-Dall-E (ResNet-50)   | 63.0       | 53.1 | 88.2.     | 39.5  | 54.5            | 66.4  | 60.8 |
| CLIP RN50 Zero-shot     | 82.1       | 55.8 | 65.9      | 41.7 | 41.6             | 85.4  | 62.1 |
| CLIP ViT-B/32 Zero-shot | 87.9      | 59.4  | 66.7     | 44.5 | 65.1              | 87.0  | 68.4 |
| CLIP ViT-B/32 Linear probe | 93.0 | 74.9  | 96.9     | 76.5  | 80.5 (full data) | 90.0 | 85.3 |

Note that directly comparing CLIP RN50 and the conventional RN50 could be unfair because CLIP ResNet-50 is not exactly the same as the conventional ResNet-50 (e.g., attention pooling, antialias pooling, ...). However, this paper uses ViT-B/32 CLIP for generating new images, and it outperforms the proposed method (except Flowers). Furthermore, if we allow access to the training datasets, then the CLIP model outperforms GIF, CLIP zero-shot with a significantly large gap (85.3 vs. 60.8 in the average accuracy). I attached CLIP ViT-B/32 linear probe results from Table 10 of the original CLIP paper [R1]. Note that the linear probe result for CIFAR100-S is made by the full training dataset, but it is known that the linear probe is less sensitive to the dataset size than the full model fine-tuning. However, I acknowledge that the comparison is not fair for CIFAR100-S (zero-shot results are fair because they are not trained with the training dataset).

Why do we need dataset expansion if CLIP zero-shot performance (or linear probe) is better than dataset expansion? If we just use CLIP zero-shot predictions, then (1) we do not need Dall-E2 or MAE decoders (2) we do not need to generate more image samples (3) we do not need to train models on the generated images. If we allow (3), the CLIP linear probe shows significant results.

I do not have CLIP results for the three medical datasets, but unless CLIP results are very poor for the medical datasets, I do not think the proposed scenario is not practical.

### Too heavy computation resources

Even if we assume that the big models trained with large-scale datasets (e.g., CLIP encoder, Dall-E2, or MAE decoder) are publicly accessible from the web, so that no extra resource is required, the proposed method still needs heavy computations to (1) generate images (2) train with x20 images. Note that the data augmentation methods usually assume the number of the total iterations is not changed, e.g., the vanilla training and Cutout augmentation for the given training dataset needs the same iterations. Following the terminology of this paper, it is common to use x1 for data augmentation. Data augmentation with xN is often called "repeated augmentation" [R2]. Usually, when we use repeated augmentation, it is not recommended to use "epochs" because a model is updated N times more using SGD optimizer, and it is known that more update brings a better performance [R3].

To summarize, the proposed method needs three additional heavy computation resources (1) training big models with large-scale datasets -- it could be ignored if we only consider image datasets, but if we consider other domains, it limits the usage of the method (2) generating images using the CLIP encoder, Dall-E2 or MAE decoder with optimized perturbations (3) the dataset expansion itself amplifies the total computational resource if we do not limit the number of iterations (i.e., x20 needs 20 times more computations than x1).

### References

[R1] Radford, Alec, et al. "Learning transferable visual models from natural language supervision." International Conference on Machine Learning. PMLR, 2021.
[R2] Touvron, Hugo, et al. "Training data-efficient image transformers & distillation through attention." International Conference on Machine Learning. PMLR, 2021.
[R3] He, Kaiming, et al. "Masked autoencoders are scalable vision learners." Proceedings of the IEEE/CVF Conference on Computer Vision and Pattern Recognition. 2022.

### Minor comments

- It looks that the bib for the COYO dataset in page 1 is broken ("Minwoo Byeon, 2022").
- It could be interesting if there exists the analysis of dropping the original images, but only using the augmented images (i.e., using only $\mathcal D_S$ in Problem statement (Section 3)
- The details of $S_{con}$ $S_{ent}$ $S_{div}$ could be more well described, e.g., showing the exact formulations using `\equation`
    - The formulation for the entropy minimization is not clear. As far as the reviewer understood, it calculates the entropy of the predicted probability (e.g., in this case, normalized cosine similarity). I cannot understand how encouraging higher prediction entropy (i.e., showing a more "uniform" prediction) helps the optimization. Is there any guess from the authors?
    - The formulation for the $S_{div}$ is not clear. How can we compute KL between features? I presume that it is achievable by optimizing the lower bound of KL divergence (as VAE)
- I wonder if the method works for larger datasets, such as CIFAR-100 (full dataset) or ImageNet-1K. Especially, I wonder about CIFAR-100 results. ImageNet-1K would not be necessary considering the scope of this paper.
- [Not necessary] It could be interesting if there are more comparisons with more advanced image augmentation methods, such as mixed sample data augmentation (e.g., Mixup, CutMix).
- [Minor] I suggest clarifying that the method is designed for "a small dataset" from the title, e.g., "Expanding Datasets With Guided Imagination for a small dataset"

**Summary Of The Paper:**

This paper proposes a Guided Imagination Framework (GIF) for generating more data samples for small datasets. The proposed model generates more image samples using two large-scale models trained with very large datasets (e.g., CLIP ViT-B/32 trained with private 400M image-text pairs and Dall-E2 decoder trained with LAION-400M). To generate new images, GIF first encodes the reference image using the CLIP ViT-B/32 encoder, and decodes a perturbed version of the encoded feature using the Dall-E2 decoder. This paper also proposes three optimization objectives based on CLIP features (both image and label text features) to seek the best perturbations for generating new samples: (1) zero-shot prediction consistency (2) entropy maximization and (3) diversity promotion. The experiments on six small natural image datasets and three small medical image datasets show that the proposed GIF shows improvements in the small dataset regime.

**Summary Of The Review:**

This paper is well-written, and the method looks fairly novel. However, the performance of the proposed method is much worse than CLIP zero-shot or linear probing, where they even use much fewer computational resources than the proposed method (no need to generate images, no need to tune the full models for x20 data samples). I think the empirical contribution and the practicalness of this paper are not enough to be accepted at the ICLR conference.

---

During the reviewer-author discussion period, the authors provide more information than the original paper. For example, the authors showed that the CLIP zero-shot is worse than the proposed method in some medical domains. However, as my following comments, I don't think the current version of the paper is not enough to be accepted at the ICLR conference. Even if we consider the additional content provided during the discussion period, the expected revision will be very significant (the main message in introduction will be revised, the main table will be revised, following explanations and the analysis will be revised); another review process will be required.

I am also not convinced by the additional experimental results. Especially, I am not positive to the additional experiments on medical domains because CLIP zero-shot shows almost random (10%) accuracy, while the quality check of the proposed generative process is done by the low-performance CLIP zero-shot. I cannot agree with the following comment
> "Although the zero-shot performance of CLIP is not good, its feature space is already relatively discriminative (given that CLIP linear-probing performance is good)"

The proposed method uses the zero-shot performance (zero-shot prediction consistency) for the "informativeness guidance". If the CLIP model has no power to discriminate the "ground-truth" images, then how can it guide whether the generated images are informative or not? Overall, I think this paper will need additional careful discussions related to my initial concerns.

---

> ### Author Response · Authors · 2022-11-18
> **Response to  Reviewer  sxmr (Part 3/3)**
>
> **Q8. Concern about the effectiveness on larger-scale datasets  like CIFAR-100.**
>
> > I wonder if the method works for larger datasets, such as CIFAR-100 (full dataset) or ImageNet-1K. Especially, I wonder about CIFAR-100 results. ImageNet-1K would not be necessary considering the scope of this paper.
>
> Thanks for  acknowledging that  our scope   is focusing on  small-data scenarios. In fact, our proposed methods can also be applied to larger datasets.  Following the suggestion, we further use our methods to expand    CIFAR100 by 5 times for model training from scratch. As shown in the following table, our GIF-DALLE leads to 8.7% accuracy improvement compared to direct training  on the original CIFAR100  dataset. Such encouraging result further verifies the effectiveness of   our methods on larger-scale datasets, and we expect to apply our methods to   even larger datasets like ImageNet in the future.   We have provided this discussion in Appendix E.7 of the revision.
>
>
> |  CIFAR100           |    Accuracy    |
> | ---------- |:---------------:|
> | Original   dataset    |    70.9+/-0.6     |
> | 5x-expanded by GIF-MAE  |  77.0+/-0.3|
> | 5x-expanded by GIF-DALLE  | **79.6+/-0.3** |
>
> ---
> **Q9. Comparisons to more advanced image augmentation methods (e.g., Mixup, CutMix).**
>
> > It could be interesting if there are more comparisons with more advanced image augmentation methods, such as mixed sample data augmentation (e.g., Mixup, CutMix).
>
> Following the suggestion, we further apply more advanced CutMix to expand CIFAR100-subset by 5 times and use the expanded dataset to  train model from scratch.  The results  in the following table further demonstrate the superiority of our method over augmentation-based expansion methods. In the revised paper, we have added this empirical discussion to Appendix E.11.
>
>
> |  CIFAR100-subset           |    Accuracy    |
> | ---------- |:---------------:|
> | Original dataset      |    35.0+/-1.7     |
> | 5x-expanded by Cutmix       |      50.7+/-0.2     |
> | 5x-expanded by GIF-DALLE  | **54.5+/-1.1**  |
>
> ---
> **Q10. Minor suggestion: the bib for the COYO dataset in page 1 is broken ("Minwoo Byeon, 2022").**
>
> > It looks that the bib for the COYO dataset in page 1 is broken ("Minwoo Byeon, 2022").
>
> Thanks    for the reminder. We have fixed it in the revision.
>
>
> ---
> **Q11. Minor suggestion: changing title.**
> >  I suggest clarifying that the method is designed for "a small dataset" from the title, e.g., "Expanding Datasets With Guided Imagination for a small dataset"
>
> Thanks a lot for the valuable suggestion. We have changed our title to "*Expanding Small-Scale Datasets with Guided Imagination*".
>
> We are happy to discuss any further questions you may have.

---

> > ### Comment · Reviewer_sxmr · 2022-11-21
> > **Response**
> >
> > Thanks for your comment and revisions. I am also sorry for the late reply.
> > I thank the authors for resolving my questions, e.g., Q3, Q4, Q5, Q8, Q9, Q10, Q11. I am not fully convinced about Q6, Q7, but they could be minor concerns that significantly change my recommendation.
> >
> > Now, I would like to start with my major concern: if we can do better with CLIP zero-shot or CLIP linear probe, then why do we have to do a dataset expansion?
> > > Why do we need dataset expansion if CLIP zero-shot performance (or linear probe) is better than dataset expansion? If we just use CLIP zero-shot predictions, then (1) we do not need Dall-E2 or MAE decoders (2) we do not need to generate more image samples (3) we do not need to train models on the generated images. If we allow (3), the CLIP linear probe shows significant results.
> >
> > In my first comment, I attached the full zero-shot and linear probe results for CLIP zero-shot and linear probe results for the experiments in this paper. The results show that only using CLIP zero-shot outperforms the proposed dataset expansion in most of the benchmarks (except Flowers) used in this paper, and the simple linear probe even outperforms the proposed method in Flowers.
> >
> > Table 11 still cannot address my concerns for three reasons
> > - (1) I argued about zero-shot performances and linear probe performances, not fine-tuning performances.
> > - (2) The proposed method is worse than CLIP zero-shot performances and CLIP linear probe performances that is much efficient than the proposed method (zero-shot does not need any additional training, linear probe only need x1 dataset training for a very small number of parameters for the linear layer)
> > - (3) [Minor] Fine-tuning performances could be affected a lot by the choice of optimization techniques. The performances could still be controversial.
> >
> > "CLIP models could be expensive for mobile terminals" is an okay motivation, but if one would like to argue this, I think the overall motivation would be focused on the comparison between CLIP-based knowledge distillation methods. But, I don't think the current version of the paper is suitable for dealing with the issue (in terms of the presentation, not in terms of the technique itself).
> >
> > I don't agree with the argument
> > > "This paper explores dataset expansion to handle real small-data scenarios, where only a small-size dataset is available without any external large-scale datasets of similar image nature. Therefore, pre-training models with CLIP on large-scale target datasets is inapplicable"
> >
> > The proposed method uses the CLIP model trained on external large-scale datasets anyway. If this argument holds, then the method should not use the CLIP model and its zero-shot capability, but it does. Or, CLIP zero-shot should be worse than the dataset expansion in some sense. I do not argue that using CLIP is problematic, but I argue that if the method uses CLIP zero-shot for the training, it should be compared with CLIP zero-shot performances. My initial review contains the official CLIP zero-shot performances, and CLIP zero-shot is better than the proposed method (which requires the additional CLIP model, DALL-E2 decoder, and the additional training procedure).
> >
> > The only advantage of this method (as far as the reviewer understood) is the capability to the different model choices, while the CLIP model is fixed and not editable. However, as far as the reviewer understood, the current presentation does not highlight this issue in the main paper but only in Appendix. I think the overall presentation should be changed to clarify that (1) the proposed method is better than CLIP zero-shot performances in some sense -- architecture selection?, (2) the proposed method should be compared to CLIP-based knowledge distillation methods -- at least for the simplest KD baseline.
> >
> > Overall, I do not agree with Q1 (as my previous comments), Q2 (it is anyway much heavier than CLIP zero-shot) which were my major concerns in my initial review.
> > > This paper is well-written, and the method looks fairly novel. However, the performance of the proposed method is much worse than CLIP zero-shot or linear probing, where they even use much fewer computational resources than the proposed method (no need to generate images, no need to tune the full models for x20 data samples). I think the empirical contribution and the practicalness of this paper are not enough to be accepted at the ICLR conference.
> >
> > Overall, because my major concerns still remain, I still recommend "3: reject" for this paper.

---

> > > ### Author Response · Authors · 2022-11-23
> > > **Follow-up response to  Reviewer  sxmr (Part 2/2)**
> > >
> > > > "CLIP models could be expensive for mobile terminals" is an okay motivation. The only advantage of this method (as far as the reviewer understood) is the capability to the different model choices, while the CLIP model is fixed and not editable.  If one would like to argue this, I think the overall motivation would be focused on the comparison between CLIP-based knowledge distillation methods.
> > >
> > > Constructive suggestion! Thanks for recognizing that *"CLIP models could be expensive for mobile terminals is an okay motivation"*, and *"the advantage of this method is the capability to the different model choices, while the CLIP model is fixed and not editable"*. It is true that  CLIP-based knowledge distillation can be used to train   various model architectures, so we further compare it with our method. As shown in the following  table, **the performance of CLIP knowledge distillation is not good enough on medical image datasets, which is the same even on natural image datasets** (cf. Table 1).  This is because CLIP knowledge distillation has two key limitations. First, distillation can only lead to limited improvement  when the performance of CLIP on the target dataset (*e.g.*, medical image domains) is not good. Second, distillation does not work well when the architectures of student and teacher models are mismatched [B,C].  Such a comparison further shows the advantage of our method that the expanded datasets  are ready for training various network architectures (Table 2), while the CLIP model architectures are fixed and not editable.
> > >
> > > [B] On the Efficacy of Knowledge Distillation.  In ICCV, 2019.
> > >
> > > [C] Does Knowledge Distillation Really Work? In NeurIPS, 2021.
> > >
> > >
> > > |  Method          |   PathMNIST   |    BreastMNIST   |    OrganSMNIST   |
> > > | ---------- |:---------------:| :---------------:| :---------------:|
> > > | Original  dataset (RN50)    |    72.4 +/-0.7     |   55.8+/-1.3   |    76.3+/-0.4     |
> > > | CLIP fine-tuning  (RN50)  |   78.4+/-0.9     |  67.2+/-2.4   |   78.9+/-0.1     |
> > > | CLIP knowledge distillation (to RN50)  | 77.3+/-1.7   |  60.2+/-1.3   | 77.4+/-0.8   |
> > > | GIF-MAE  (RN50)  |  82.0+/-0.7 |   73.3+/-1.3      |     80.6+/-0.6    |
> > > | GIF-DALLE  (RN50)   | 84.4+/-0.3  |   76.6+/-1.4      |    80.5+/-0.2  |
> > >
> > > ---
> > > > It is much computationally heavier than CLIP zero-shot.
> > >
> > >
> > > We agree that the computational efficiency of the proposed method is not high now, but as mentioned in our previous response,  **computational efficiency is not the focus of this paper**.  This work  explores   the possibility of using recent generative models for dataset expansion, and extensive experiments have demonstrated  dataset expansion as a promising future direction. We believe   **exploring new promising   research directions is valuable to the academic community, while it is unnecessary to resolve and end the research direction at the beginning**. Hence, we regard the computational efficiency of dataset expansion as an important research issue that can be improved in the future.
> > >
> > >
> > > ---
> > >
> > > Thank  the reviewer  again for providing highly constructive advice and being willing to conduct an in-depth discussion.  Following the suggestion,  we have  further revised the paper by incorporating the above discussion into Section 5.3 and Appendix E.6 to clarify the advantages of dataset expansion over directly using/transferring CLIP models (although we cannot update a new revision now). We hope this response can  address your main concerns   and look forward to discussing any  remaining questions.

---

> > > ### Author Response · Authors · 2022-11-23
> > > **Follow-up response to  Reviewer  sxmr (Part 1/2)**
> > >
> > >
> > > Great discussion! We highly appreciate the reviewer for patiently reading our response and providing in-depth discussions, which is very helpful for improving the quality of our paper. We next address the main concerns as follows.
> > >
> > > ---
> > >
> > > >  I attached the full zero-shot and linear probe results for CLIP zero-shot and linear probe results for the experiments in this paper. The results show that only using CLIP zero-shot outperforms the proposed dataset expansion in most of the benchmarks (except Flowers) used in this paper, and the simple linear probe even outperforms the proposed method in Flowers. Table 11  still cannot address my concerns. I argued about zero-shot performances and linear probe performances, not fine-tuning performances.
> > >
> > > We see your point and also thank the reviewer for providing the zero-shot and linear-probing results   of CLIP on natural image datasets. As  mentioned in our previous response, we acknowledge that CLIP performs well on natural image datasets. We are not attacking this, but it is worth noting that the applicability of CLIP  to various other domains like medical image datasets is limited. In the initial response, we illustrate this by providing the fine-tuning performance of CLIP, since  **its fine-tuning performance  is  much better than its zero-shot prediction and linear probing on medical image datasets**. As shown in the following table, it is more evident that CLIP's zero-shot prediction and linear probing do not work well on three medical datasets. The poor zero-shot performance is attributed to that the CLIP models,  trained on natural images, have limited representation abilities on medical images and poor zero-shot discriminability  of medical categories (*e.g.*, 'adipose', 'lymphocytes' and 'cancer-associated stroma' in PathMNIST data). The limited linear-probing or fine-tuning performance is  because,   when the pre-trained datasets are highly different from the target datasets, the pre-training weights   do not significantly help performance compared to training from scratch [A]. Such a new result further shows the better applicability of our method   to the scenarios of various image domains.
> > >
> > > We further explain why we can use CLIP to guide dataset expansion in those domains. Although the transfer performance of CLIP is not significantly good, its feature space is already relatively discriminative. The boundary of the corresponding classifier thus   helps to measure   prediction entropy, and acts as an  anchor point  to enforce  class semantic consistency  of   expanded images. Hence,   CLIP can be used in our method to conduct guided imagination for various  image domains.
> > >
> > >  [A] Transfusion: Understanding transfer learning for medical imaging. In NeurIPS, 2019.
> > >
> > >
> > > |  Method          |   PathMNIST   |    BreastMNIST   |    OrganSMNIST   |
> > > | ---------- |:---------------:| :---------------:| :---------------:|
> > > | Original  dataset (RN50)    |    72.4 +/-0.7     |   55.8+/-1.3   |    76.3+/-0.4     |
> > > | CLIP   zero-shot  (RN50)    |   10.7   |  51.8   |  7.7     |
> > > | CLIP   linear probe  (RN50)  |   74.3+/-0.1   |  60.0+/-2.9  |    64.9+/-0.2 |
> > > | CLIP fine-tuning  (RN50)  |   78.4+/-0.9     |  67.2+/-2.4   |   78.9+/-0.1     |
> > > | GIF-MAE  (RN50)  |  82.0+/-0.7 |   73.3+/-1.3      |     80.6+/-0.6    |
> > > | GIF-DALLE  (RN50)   | 84.4+/-0.3  |   76.6+/-1.4      |    80.5+/-0.2  |
> > > ---
> > >
> > > > I don't agree with the argument of "This paper explores dataset expansion to handle real small-data scenarios, where only a small-size dataset is available without any external large-scale datasets of similar image nature. Therefore, pre-training models with CLIP on large-scale target datasets is inapplicable". The proposed method uses the CLIP model trained on external large-scale datasets anyway. If this argument holds, then the method should not use the CLIP model and its zero-shot capability, but it does
> > >
> > > We are not arguing that we cannot access all large datasets, but the ones with **similar data nature to the target dataset in real small-data  scenarios**, such as large-scale  pathological medical data. In those scenarios, re-training models with CLIP on large-scale **target datasets** is inapplicable, so the aforementioned  limitation of CLIP models     on various image domains cannot be resolved by conducting CLIP  pre-training on these domains.

---

> > > ### Author Response · Authors · 2022-11-25
> > > **Summary of our response**
> > >
> > > Thanks again for the constructive comments. We would like to summarize our response further. According to the comments, the main concerns of the reviewer are: (1) the superiority of our dataset expansion method over CLIP; (2) the computational costs of our method.
> > >
> > > In our response, we address the first concern by providing more empirical comparisons between our method and CLIP. There are two main advantages of our methods.
> > >
> > > - **(1) Our dataset expansion method has better applicability to the scenarios of different image domains than CLIP**. Specifically, our GIF-DALLE performs significantly better than CLIP's zero-shot, linear-probing, fine-tuning performance on medical domains (*i.e.*, obtaining at least 5.7% accuracy gains on average over 3 medical datasets).
> > >
> > > - **(2) Our method can provide expanded datasets for training various network architectures, while the  CLIP model architectures are fixed and may be unusable in real scenarios with device hardware constraints**. Note that simply conducting distillation with CLIP performs much worse than our method. For example, GIF-DALLE outperforms CLIP distillation by 27.2% accuracy gains on average over 6 natural image datasets and by 8.9% accuracy gains on average over 3 medical datasets. Please refer to the full response for details.
> > >
> > >
> > > For the second concern, we would like to highlight that our expanded datasets are reusable and can train various DNN models without re-expansion, which helps to **reduce significantly the cost of human data collection and annotation in real applications**. Extensive experiments have shown the empirical superiority of the proposed method over CLIP and other baselines. Considering that computational efficiency is not the focus of this paper, we leave it to the future.
> > >
> > >
> > >
> > > We sincerely appreciate the in-depth discussion. We believe that our point-by-point response can address the two concerns. Do you have any remaining concerns that we can resolve?

---

> > > > ### Comment · Reviewer_sxmr · 2022-12-05
> > > > **Response**
> > > >
> > > > Thanks for your additional comments. I am still skeptical about revising my score due to the following reasons:
> > > >
> > > > - Even though the authors newly attached experimental results for three medical datasets, this paper shows its effectiveness in natural images (Table 1). It should need a heavy revision to revise the overall experimental results and corresponding writings. If the main table will be retained without attaching CLIP zero-shot results and corresponding discussions, I am not able to recommend acceptance. In the case of when CLIP zero-shot results and linear probe results are added to the current main table, then I will maintain my score because of my initial concern.
> > > >     - There could be a minor concern in terms of the method in this case. If CLIP zero-shot only shows 10% accuracy (=random choice), then it may imply that the "informativeness guidance" could be useless or harmful, because the selected samples will be guided by the poor CLIP zero-shot performance. In my opinion, it needs more verification and discussion.
> > > > - I partially agree with the architecture issue, but in that case, the main experimental results should be based on much smaller architectures, such as MobileNet. It is partially addressed in Table 2, but I think that the results should be expanded to the whole datasets, rather than one dataset (Cars).
> > > > - In my opinion, the first result should be the medical dataset results (compared with CLIP zero-shot and linear probe) based on major architectures (RN-50 or ViT). The second result should be the case of different architecture choices. The second result should be compared on the full dataset used in the first result. I also think that the verification and justification of the choice of "informativeness guidance" when CLIP zero-shot performance is nearly "random" (10% or 7%).
> > > >
> > > > Compared to my initial review, I am now less skeptical about the proposed method. It could be valuable in a specific scenario (1) when CLIP zero-shot is not working, (2) when one needs different architectures than CLIP image encoders, and (3) when naive fine-tuning of ImageNet or CLIP pre-trained model does not work well. However, I am still skeptical about recommending acceptance for this paper because the current version of the paper does not argue that the proposed method is for the specific scenario, but the proposed method is for a general scenario, even when CLIP zero-shot significantly outperforms the proposed method. I do not think the modification is small enough, but it needs a heavy modification, including the overall paper structure and experimental results. I encourage the authors to deal with my concerns and comments in a future revision.

---

> > > > > ### Author Response · Authors · 2022-12-06
> > > > > **Follow-up response to Reviewer sxmr**
> > > > >
> > > > > Thanks for the follow-up comment. We are glad to see that the reviewer is more convinced of our method and see the value of our paper. We address the remaining concerns as follows.
> > > > >
> > > > > ---
> > > > >
> > > > > > Compared to my initial review, I am now less skeptical about the proposed method. It could be valuable in specific scenario. I am still skeptical about recommending acceptance for this paper because the current version of the paper does not argue that the proposed method is for the specific scenario, but the proposed method is for a general scenario, even when CLIP zero-shot significantly outperforms the proposed method.
> > > > >
> > > > > This paper explores a **novel dataset expansion setting for general image domains**, and has shown huge potential of using recent generative models for dataset expansion on both medical images (12.3% accuracy gain on average over 3   datasets) and natural images (29.9% accuracy gain on average over 6   datasets). Please note that, based on the results provided by your original comment,  CLIP RN50 zero-shot performs only slightly better than the RN50 trained by our method on average over 6 natural datasets (60.8 [GIF-DALLE R50] vs 62.1 [CLIP R50 zero-shot]).  **Such a slight performance gap is tolerated, since CLIP RN50 is trained on a significantly larger dataset (400 million data) than ours (at most 60,000 data)**. More critically, when further considering medical datasets, our method significantly outperforms CLIP zero-shot by 18.2% accuracy on average over 9 datasets (c.f. the following table). In addition,  **our expanded datasets are more flexible than CLIP models to train various model architectures**, so our method is more applicable to real application scenarios (e.g., with hardware constraints). We believe that, **as a pioneering dataset expansion work, our method is not required to completely resolve the task with overwhelming performance**. We expect the performance of dataset expansion on natural images can be further improved in future research.
> > > > >
> > > > >
> > > > > |  Method          |   PathMNIST |	BreastMNIST	|OrganSMNIST 	 |  Caltech	| Cars   |    	Flowers  |    	DTD	  |    C100-S  |    	Pets   | Average |
> > > > > | ---------- | :---------------:|  :---------------:| :---------------:| :---------------:| :---------------:| :---------------:| :---------------:| :---------------:| :---------------:| :---------------:|
> > > > > | CLIP   zero-shot  (RN50)    | 10.7 |	51.8	 | 7.7   | 82.1 |  55.8	 | 65.9 | 	41.7 | 	41.6	 | 85.4   |  49.2 |
> > > > > | GIF-DALLE (RN50 ) |	 84.4	|  76.6	|  80.5| 63.0|	53.1	|88.2 | 39.5	|54.5|	66.4 | 67.4 |
> > > > >
> > > > >  ---
> > > > >
> > > > > > In my opinion, the first result should be the medical dataset results. The second result should be the case of different architecture choices. The second result should be compared on the full dataset used in the first result.
> > > > >
> > > > > Thanks for the suggestion, following which we will put the experiments of medical image datasets to Section 5.1 as the first result and illustrate the method effectiveness on various architectures as the second result. For the latter one, in our preliminary studies, **the effectiveness of our method on various architectures is consistent among all datasets**. We will provide the results **on other datasets** as soon as possible. Moreover, since we cannot update the paper in the current stage, we will show the modification in the camera-ready.
> > > > >
> > > > > ---
> > > > >
> > > > > > There could be a minor concern. It needs  justification of the choice of “informativeness guidance” when CLIP zero-shot performance is nearly “random” (10% or 7%).
> > > > >
> > > > > We further compare GIF-MAE results with zero-shot CLIP and with fine-tuned  CLIP based on OrganSMNIST, on which CLIP zero-shot performance is not good. The following table shows that **GIF-MAE with zero-shot CLIP performs only comparably to  GIF-MAE with fine-tuned CLIP**. This result reflects that the CLIP zero-shot classifier is enough to provide sound guidance. Although the zero-shot performance of CLIP is not good, its feature space is already relatively discriminative  (given that CLIP linear-probing performance is good). The boundary of the corresponding classifier thus can act as an anchor point to enforce the class semantic of the expanded image to remain close to the seed image in the feature space, and helps to measure entropy for increasing classification difficulties. Therefore, zero-shot  CLIP can be used in our method to conduct guided imagination for medical image domains.  **More importantly, we can also fine-tune the pre-trained CLIP on medical image datasets for dataset expansion, if it is required in practice.**
> > > > >
> > > > >
> > > > >
> > > > > |  Method          |      OrganSMNIST   |
> > > > > | ---------- | :---------------:|
> > > > > | CLIP   zero-shot      |   7.7     |
> > > > > | CLIP   linear probe   |      64.9+/-0.2 |
> > > > > | CLIP fine-tuning    |   78.9+/-0.1     |
> > > > > | GIF-MAE with fine-tuned  CLIP   |   80.1+/-0.3   |
> > > > > | GIF-MAE with zero-shot   CLIP  |   80.6+/-0.6    |
> > > > >
> > > > > ---
> > > > > We sincerely appreciate the in-depth discussion.

---

> ### Author Response · Authors · 2022-11-18
> **Response to  Reviewer  sxmr (Part 2/3)**
>
> **Q3. Concern about the evaluation manner regarding training epochs.**
>
> > Note that the data augmentation methods usually assume the number of the total iterations is not changed, e.g., the vanilla training and Cutout augmentation for the given training dataset needs the same iterations. It is not recommended to use "epochs" because a model is updated N times more using SGD optimizer, and it is known that more update brings a better performance.
>
> This paper studies a new task of dataset expansion, where the key goal is to explore how to expand datasets better instead of how to augment data better.   To **fairly evaluate the expansion effectiveness** of different methods,  the original small datasets are **expanded by the same ratios**, followed by training     models from scratch on   the expanded dataset with **the same   number of epochs and same data pre-processing**. In this way, the models are trained with the same  number of  update steps, so   all expansion methods are fairly compared.   We have clarified this in Appendix C.3 of the revision.
>
> ---
> **Q4. Comparison to training models with only expanded images.**
> > It could be interesting if there exists the analysis of dropping the original images, but only using the augmented images.
>
> Training   models with only expanded images is interesting. Following the advice, we only use the expanded images by GIF-DALLE on   CIFAR100-subset   to train ResNet-50 from scratch, and compare it to the model trained on real images of CIFAR100-subset. As shown in the following table,  the model trained with our synthetic images performs comparably to the model trained with real images. This result further   verifies  the effectiveness of our explored dataset expansion method. Moreover,   the model trained with the full expanded dataset performs much better than that trained with only  the original dataset or  with only  the generated images. That is, the generated images are not a simple repetition of the original dataset, but bring new information to the expanded dataset for model training. This further shows   that using synthetic images for model training is a promising direction. We expect that our work on dataset expansion can inspire more studies to explore this direction in the future. In the revised paper, we have added this discussion to Appendix E.10.
>
> |  CIFAR100-subset            |    Accuracy    |
> | ---------- |:---------------:|
> | Original  dataset       |    **35.0+/-1.7**     |
> | Expanded  dataset by GIF-DALLE    |      54.5+/-1.1     |
> | Only expanded images by GIF-DALLE    |  **35.2+/-1.3** |
>
> ---
> **Q5. The details of three guidance   could be more well described with exact equations.**
>
> Following the advice,  we have further provided the exact equations for all the proposed guidance  in Appendix C.1 of the revised paper.
>
> ---
> **Q6. Clarification on    entropy maximization.**
>
> Information entropy measures the informativeness of a message in information theory. In this paper, entropy maximization $\mathcal{S}_{ent}$ (cf. Eq. 4 in Appendix C.1) encourages  the   synthetic image   to have  larger information entropy    than that of  the seed image with respect to CLIP zero-shot predictions. As a result,  the   synthetic image  is more challenging for classification than the seed image, and thus brings new information to the expanded dataset  for model training (as mentioned in  Section 3.2). The ablation study of Table 8 shows that such a guidance  is able to bring performance gain for dataset expansion.
>
>
> ---
> **Q7. Clarification on the implementaton of diversity promotion.**
>
> As mentioned at the  bottom of Page 5, the diversity of latent features is measured by  $\mathcal{S}_{div} = \text{KL}(f'; \bar{f})$, where  $f'$  denotes the current  perturbed latent feature  and $\bar{f}$ indicates the  mean over  the $K$ perturbed latent features of this seed sample. Here, we measure   the dissimilarity of two feature   vectors by applying the softmax function to the latent features, and then measuring the KL divergence between the resulting probability vectors. Please note that employing  KL divergence in the feature space  has been widely used in machine learning studies [A-D].  We further clarify this in Appendix C.1 of   the revised paper.
>
>
> [A] Simultaneous Similarity-based Self-Distillation for Deep Metric Learning. ICML, 2021.
>
> [B] Unsupervised deep embedding for clustering analysis. ICML, 2016.
>
> [C] Improved Deep Embedded Clustering with Local Structure Preservation. IJCAI, 2017.
>
> [D] Image Matting with KL-Divergence Based Sparse Sampling. ICCV, 2015.

---

> ### Author Response · Authors · 2022-11-18
> **Response to  Reviewer  sxmr (Part 1/3)**
>
> Thanks for the constructive comments. We are glad to see that our thorough empirical analysis and good writing  are appreciated.  We  address the main concerns point by point as follows.
>
> ---
> **Q1. Concern about the performance of dataset expansion  compared to CLIP.**
>
> This paper explores dataset expansion to handle real small-data scenarios, where   only a small-size dataset is available without any external large-scale  datasets of similar image nature. Therefore, pre-training  models with CLIP   on large-scale target datasets  is inapplicable. In this work, we resort to publicly  available CLIP models for dataset expansion. Compared to directly transferring CLIP models, our  dataset expansion  is   a necessarily new paradigm for the following two key reasons.
>
> - **Our dataset expansion method has better applicability to  the scenarios of different image domains**. Although CLIP has strong  transfer performance on some natural image datasets, its transfer performance to other domains like medical image datasets is limited. Here, we further report the fine-tuning performance of CLIP-trained ResNet-50 model on three medical datasets. As shown in the following table,  transferring   the CLIP model   only leads to  limited performance gains, which are significantly worse than our dataset expansion. The reason is that,   when the pre-trained datasets are highly different from the target dataset, the pre-training and fine-tuning scheme does not significantly help performance [A]. In contrast, our GIF framework can generate images of similar nature as the target dataset for expansion, and thus  is more beneficial to real applications of various image domains.
>
>   [A] Transfusion: Understanding transfer learning for medical imaging. In NeurIPS, 2019.
>
>
> |  Method          |   PathMNIST   |    BreastMNIST   |    OrganSMNIST   |
> | ---------- |:---------------:| :---------------:| :---------------:|
> | Original  dataset     |    72.4 +/-0.7     |   55.8+/-1.3   |    76.3+/-0.4     |
> | Fine-tuned CLIP   |   78.4+/-0.9     |  67.2+/-2.4   |   78.9+/-0.1     |
> | GIF-MAE  |  82.0+/-0.7 |   73.3+/-1.3      |     80.6+/-0.6    |
> | GIF-DALLE (fine-tuned DALL-E)  | 84.4+/-0.3  |   76.6+/-1.4      |    80.5+/-0.2  |
>
>
>
>
> -  **Our dataset expansion  can provide expanded datasets ready for training various network architectures**. In some real application scenarios like mobile terminals, the supportable  model size  is very limited due to the constraints of   device hardware. However, the publicly available checkpoints provided by CLIP are only ResNet50, ViT-B/32 or even larger models, which may not  be allowed to use in those scenarios.  By contrast, the expanded dataset by our method can be directly used to train various model architectures  (cf. Table 2), and thus is more   applicable to the  application scenarios with hardware constraints. Also, once the  datasets are expanded, they  can   be released to the public to facilitate future studies.
>
> Please note that the goal of  this paper  is  **to show the huge potential of dataset expansion as a promising future  direction  instead of  completely resolving it**. We expect  that the performance of dataset expansion can be further improved in   future research.  We have further clarified this in the Appendix E.6 of the revised paper.
>
> ---
>
> **Q2. Concern about computational costs of dataset expansion.**
>
> Please note that automatic dataset expansion can help to reduce the time and cost of human data collection/annotation for real small-data applications. In this work, to better expand    datasets automatically,  we explore   the possibility of using recent generative models for dataset expansion, while **computational efficiency is not the focus of this paper**. In fact,  we can resort to numerous  publicly available generative models for dataset expansion without training them from scratch.  Our GIF framework is general, so it can   work with other generative models with faster inference. Moreover, the datasets are reusable once expanded and can be directly applied to train various DNN models without  re-expansion. These help to reduce the computational costs of our dataset expansion method. How to conduct more resource-efficient dataset expansion is an important and non-trivial open question, and we leave it to the future.

---

### Official Review · Reviewer_nKUi · 2022-10-23

**Confidence:** 4
**Correctness:** 3
**Technical Novelty And Significance:** 3
**Empirical Novelty And Significance:** 3
**Recommendation:** 6

**Clarity, Quality, Novelty And Reproducibility:**

Overall, the paper is well-written. The method and experiments are clearly described.
As mentioned before, the GIF-MAE variant could be better integrated and additional discussions on limitations would be useful in the main paper.

As a related work that also leverages latents of deep generative networks to build augmentations, the authors could mention:

Generative Models as a Data Source for Multiview Representation Learning. ICLR 2022.
Ali Jahanian, Xavier Puig, Yonglong Tian, Phillip Isola

**Strength And Weaknesses:**

Strengths:
- The paper tackles a very relevant problem of training in a low-data regime. Using generative models for this purpose is an intuitive solution that shows promises but has so far been struggling to obtain significant success. The paper is a step in that direction.
- The diverse set of experiments provides good insights into the method. I found the results on medical images and the corresponding discussion in Appendix E regarding the limitations of GIF-DALLE especially important, and believe there should be more discussions in the main paper on that subject.

Weaknesses:
- It is not clear how GIF-MAE fits in the narrative.
  - In most sections the main focus is GIF-DALLE and the MAE variant is barely mentioned, almost like an afterthought. In the experiments, however, it takes a very significant portion.
The authors might want to fix this imbalance. It could be done by integrating GIF-MAE earlier in the motivation, or instead by treating it even more entirely as a side contribution in the later sections.
  - The sentence in the introduction "Considering that DALL-E2 and MAE (He et al., 2022) have been shown to be powerful in generating and reconstructing images, we explore their use as prior generative models for imagination in this work." seems particularly misleading. Can the authors clarify if they consider MAE a powerful image generator, and if so, in which sense?
- The framing of the results is useful but seems artificial. As such the presented results are insufficient to draw conclusions regarding the usefulness of the method in practice.
  - Comparing their methods against standard data augmentations with a fixed number of samples is fine, but it would be useful to have the performance of the baselines with a virtually infinite number of augmented samples.
Indeed, a key difference between the two approaches is that, in practice, data generation has significant memory and a computation footprint while it should be very easy to add infinite standard data augmentation in the training pipeline with little cost.
  - This comparison might also not be fair as the GIF-DALLE has had access to Laion-400M. Another interesting baseline would have been to simply automatically select and label additional unlabelled data using CLIP embeddings. This could also serve as some sort of ablation study to check if the generator is actually useful, or if directly using the raw dataset with CLIP pseudo-labeling is enough.

Question:
- It is mentioned that GIF-DALLE generates images of size 64x64 (Appendix C.1) while the downstream models use inputs of size 224x224 (Appendix C.3). These numbers seem off considering the shown samples. Can the authors confirm these numbers are used for all datasets?  If so, how exactly are the outputs of GIF-DALLE upsampled and processed, especially with regard to the random cropping mentioned in C.3?

**Summary Of The Paper:**

The paper proposes to tackle a new task of dataset expansion, which is to automatically find new labeled data to augment an existing dataset.
To do so, they propose to generate them using a generative model (DALL-E2) using CLIP guidance.
The idea is that using DALL-E2 would be able to generate variations around existing samples, and CLIP can be used to guide the generated samples to only generate useful data.
In addition to DALL-E2, they show they can also use Masked Auto-Encoders (MAE) as a backbone, with the same CLIP guidance.
They compare their method of expansion against standard data-augmentation techniques on a set of small natural image datasets, as well as medical image datasets.

**Summary Of The Review:**

The paper tackles an interesting problem in a novel way, showing promising results. It contains a lot of information, shedding light on the capabilities of current generative models and text embedding to synthesize training data.

My two main concerns are 1) about the integration of the GIF-MAE model in the narrative of the paper and 2) that the current experiments are not sufficient to be convinced about the usefulness of the method in practice. Additionally, I would also like some discussions regarding the limitations of the method in the main paper.

---

> ### Author Response · Authors · 2022-11-18
> **Response to  Reviewer nKUi (Part 2/2)**
>
> **Q3. Comparison to picking related samples from larger datasets.**
>
>
>
> Picking and labeling data from larger image datasets with CLIP is an interesting  idea for dataset expansion.  However, such a solution is limited in   real applications, since   a   large-scale   related dataset may be  unavailable in many image domains. Moreover, selecting data from different image domains (*e.g.*, from natural images to medical images) is unhelpful  for dataset expansion.
>
>
> Even so, we also follow the suggestion to provide this baseline on CIFAR100-subset, by using CLIP to select and annotate related images from ImageNet to expand CIFAR100-subset by 5 times. Specifically, we scan over all ImageNet images and use CLIP to predict them to the class  of CIFAR100-subset. We select the samples with the highest prediction probability higher than 0.1 and expand each class by 5x. As shown in the following table, the idea of picking related images from ImageNet  makes sense, but performs worse than our proposed method. This result further demonstrates the  effectiveness and superiority of our method. In addition, how to  better transfer large-scale datasets  to expand small datasets is an interesting open question, and   we expect to explore it in the future. This discussion has been added to Appendix E.9 of the revision.
>
>
>
> |  CIFAR100-subset           |    Accuracy    |
> | ---------- |:---------------:|
> | Original   dataset    |    35.0+/-1.7     |
> | 5x-expanded by picking data from ImageNet with CLIP       |      50.9 +/-1.1  |
> | 5x-expanded by GIF-DALLE  | **54.5+/-1.1**  |
>
> ---
> **Q4. Clarification on image resolutions.**
>
> The values are accurate and used for all datasets. The resolution of the generated images by GID-DALLE is 64x64, and  in downstream training   we    resize the images to 224x224 through  bicubic upsampling (cf. Appendix C.3).  Moreover, for clear visualization of the generated images in this paper,   we use the  the super-resolution model of DALL-E2 to further upsample the generated images  to  256x256 resolution and format them manually.
>
> ---
> **Q5. Related work of "Generative Models as a Data Source for Multiview Representation Learning (ICLR 22)".**
>
> Thanks for  the recommendation. The mentioned study explores the possibility of using  GAN models to generate images for contrastive model pre-training,   also showing that training models with synthetic data is a promising direction.  However, as the generated images are without labels, they cannot be directly used for expanding small datasets.  In contrast, our proposed  dataset expansion methods can expand a real small dataset to a larger labeled one in a  fully automatic manner. Moreover, our work also    reveals the possibility and huge potential of   recent   generative diffusion models (like DALL-E2) to generate synthetic images for model training. We have cited and mentioned  the recommended paper in the revised paper.
>
>
> Thanks again for your constructive comments. We welcome to discuss any further questions you may have.

---

> ### Author Response · Authors · 2022-11-18
> **Response to  Reviewer nKUi (Part 1/2)**
>
> Thanks a lot for the valuable comments. We are glad to see that the importance of the explored task and the insights provided by our work are appreciated. We address the main concerns point by point as follows.
>
> ---
>
> **Q1. MAE is too little described in the method  section.  Clarification of whether the authors consider MAE as image generator.**
>
>
> In this work, our main contribution is the guided imagination framework (GIF) inspired by human learning with imagination,  so the story of this paper revolves around GIF.   To facilitate understanding, we use DALL-E2 as an example to illustrate GIF  rather than emphasizing GIF-DALLE. Note that GIF is a general framework that can work with various generative and reconstruction models, where   DALL-E2 and MAE are two use cases. The detailed implementation of GIF-MAE was   provided  in Appendix C.2 of the paper.
>
> The model trained by MAE is not a generative model. MAE was initially designed for self-supervised representation learning through masked  reconstruction, so it can be regarded as a reconstruction model by setting the masking ratio to zero in inference. Please note that the ability of image reconstruction can also be used for image generation with careful design. Thanks to the  strong reconstruction ability of MAE, we   explore its use as a prior model for data imagination. In the revision, we have revised  the  sentence accordingly and  further mentioned MAE in  Abstract.
>
>
>
> ---
> **Q2. Concern about the evaluation method.  Comparison to infinite data augmentation**.
>
> In this work, we are exploring a new task of dataset expansion, where the key goal is to explore how to expand datasets better instead of how to augment data better.  To **fairly evaluate the expansion effectiveness** of different methods,  the original small datasets are **expanded by the same ratios**, followed by training     models from scratch on   the expanded dataset with **the same   number of epochs and same data pre-processing**. In this way, the models are trained with the same  number of  update steps, so   all expansion methods are fairly compared. Following this evaluation manner, we have demonstrated that our proposed methods are more effective and efficient in expanding small datasets.
>
>
>
> Here, we also follow the advice and further evaluate the performance of infinite data augmentation  on  CIFAR100-subset. Specifically, based on RandAugment, we train ResNet-50 using infinite online augmentation for varying numbers of epochs from 100 to 700.  As shown in the following table, using RandAugment to train models for more epochs  leads to better performance, but gradually converges   (around 51% accuracy at 500 epochs) and  keeps fluctuating afterward. By contrast, our GIF-DALLE can achieve better performance when only training 100 epochs, which further demonstrates the effectiveness of our method in generating informative synthetic data for model training.  We have provided this discussion in Appendix E.8 of the revised paper.
>
> |  CIFAR100-subset     |    Epochs   |      Accuracy    |
> | ---------- |  :---------------:|   :---------------:|
> | Standard training |   100 |  35.0+/-1.7     |
> |  Training with RandAugment    | 100 |  39.6+/-2.5  |
> | Training with RandAugment     | 200 |   46.9+/-0.9  |
> | Training with RandAugment     | 300 |    48.1+/-0.6  |
> | Training with RandAugment     | 400 |   49.6+/-0.4  |
> |  Training with RandAugment         | 500 |     **51.3+/-0.3** |
> |  Training with RandAugment      | 600 |   **51.1 +/-0.3**  |
> |  Training with RandAugment  | 700 |   **50.6 +/-1.1**  |
> | 5x-expanded by GIF-DALLE    | 100 |     **54.5+/-1.1**  |

---

### Official Review · Reviewer_DLM1 · 2022-10-24

**Confidence:** 4
**Clarity, Quality, Novelty And Reproducibility:** The quality, clarity and originality …
**Correctness:** 3
**Technical Novelty And Significance:** 4
**Empirical Novelty And Significance:** 3
**Recommendation:** 8

**Strength And Weaknesses:**

Pros:
+ The setting of this paper is new and interesting. A similar idea has been applied in the GAN models, which seems not to be working. The latest progress in diffusion models is significant. It has shown a great enhancement in the quality of synthetic images. Applying such generative models to help discriminative tasks is a natural and straightforward idea. To the best of my knowledge, this is a pioneering work in this area.

+ This paper has achieved very promising results over many benchmarks, with a 29.9% accuracy gain on average over six natural image datasets and a 10.4% accuracy gain on average over three medical ones. Such an improvement is significant compared with non-parametric data augmentations.

+ With the help of three criteria as guidance, there is a further accuracy gain due to the increase of new information in generated data. The proposed disturbance is simple and effective.

Cons:
- One of my major concerns is that all the evaluations are conducted over small datasets where MAE or DALL-E should easily cover such domains. I am not very surprised by the improvement in these datasets. I am wondering if this will be helpful for other tasks/domains on a larger scale.

- There is no sufficient theoretical analysis in this paper. The whole paper is based on empirical studies with solid experiments whose quality can be further improved with more discussion of analysis in the theoretical aspect.

- Despite the expansion efficiency of GIF claimed in Sec. 5.1, I am doubtful about the time and cost efficiency due to the low speed and high computation of the diffusion model in inference.

Questions and Other Concerns:
1. I am wondering what is the open-source plan of this paper since the DALL-E is not an open-sourced model.

2. It would be good if the authors shared the time and computational costs needed for dataset expansion.

3. The images generated by MAE seem to have bad visual quality. How to explain the benefits of such noisy images for augmentation? Is it any possibility that they could be harmful to accuracy if the image quality is pretty bad and why?


**Summary Of The Paper:**

This paper presents a Guided Imagination Framework (GIF) for data augmentation. The original small-set dataset can be expanded by generating new images from powerful generative models such as DALL-E or MAE. Compared with the non-parametric data augmentations, such generative-models-guided ones bring extra new information due to the nature of the models themselves. Diversity and fidelity are the two crucial factors for success. This paper proposed three criteria as guidance: prediction consistency, entropy maximization, and diversity promotion. GIF is verified on multiple natural or medical image datasets with a stable boost compared with the baselines.

**Summary Of The Review:**

Despite many concerns about this paper, I am still positive due to its novelty and the promising results in many small datasets.

---

> ### Author Response · Authors · 2022-11-18
> **Response to  Reviewer DLM1 (Part 2/2)**
>
> **Q3. Concern on computational efficiency of dataset expansion.**
>
>
> Please note that automatic dataset expansion can help to reduce the time and cost of human data collection/annotation for real small-data applications. In this work, to better expand    datasets automatically,  we explore   the possibility of using recent generative models for dataset expansion, while computational efficiency is not the focus of this paper.   We agree that the current computational costs of diffusion models like DALL-E are   high (as shown in the following table), but we expect    the efficiency issue of diffusion models can be accelerated in future research [B,C,D] . Moreover, please note that our GIF framework is general, so it can also work with other generative models with faster inference.
>
> | Method          | 5x-expansion  time  per sample  | Params (M)  |
> | ---------- |:---------------:|   :---------------:|
> | GIF-DALLE       |    29s  on one  V100 GPU    | 2339.04M|
> | GIF-MAE        |       0.04s  on one  V100 GPU       |  480.82 M|
>
> [B] Progressive distillation for fast sampling of diffusion models. In ICLR, 2022.
>
> [C] Efficient Spatially Sparse Inference for Conditional GANs and Diffusion Models. In NeurIPS, 2022.
>
> [D] DPM-Solver: A Fast ODE Solver for Diffusion  Probabilistic Model Sampling in Around 10 Steps. In NeurIPS, 2022.
>
>
>
> ---
> **Q4. I am wondering what is the open-source plan**.
>
> Our proposed GIF framework is general and can work with various generative models besides DALL-E2. We will   release our source code and the expanded datasets.
>
> ---
> **Q5. Concern about the visual quality of MAE-generated images.**
>
> The fundamental reason for the effectiveness of  GIF-MAE is that it can generate style-diversified images  while keeping the content unchanged. This ie achieved by conducting channel-level perturbation over latent features (rather than  pixel-level   perturbation) in our method (please refer to Appendix B for detailed analysis and Section 5.2 for empirical benefits). Thanks to this characteristic, the generated images by GIF-MAE (cf. Appendix F.1) are style-diversified while keeping the same class semantics, thus bringing  performance gains for model training (cf. Table 1).
>
>  In fact, if the expanded images are generated  without any guidance, highly low-quality synthetic images may cause class semantic shifts and bring performance degradation. However, as our proposed GIF is designed based on three guidances (*i.e.*,   prediction consistency, entropy maximization  and  diversity promotion) for dataset expansion, the problems like  class semantic shifts are alleviated. This is  supported by the ablation study of Table 8 (cf. Appendix E.2), where the expansion performance is decreased if we do not use those guidance.
>
> We are glad to discuss any further questions you may have.

---

> ### Author Response · Authors · 2022-11-18
> **Response to  Reviewer DLM1 (Part 1/2)**
>
> Thanks for the encouraging comments on our paper, particularly for recognizing the value of the explored problem  and our proposed method (*"The setting of this paper is new and interesting. This is a pioneering work in this area."*). We hope that our exploration can inspire future research to  address the data limitation issue from a new perspective of dataset expansion. We   address   the concerns as follows.
>
> ---
>
> **Q1. All the evaluations are conducted over small datasets where MAE or DALL-E should easily cover such domains. I am wondering if this will be helpful for other tasks/domains on a larger scale.**
>
>
> The  goal  of this paper is to address **the data limitation issue in real small-data scenarios**, and we have shown the effectiveness of our method on  both natural and **medical image domains**. Please note that the medical   images are not covered by the training set of MAE, but  our method also works well on medical domains (cf. Table 3).
>
> Although expanding larger-scale datasets is not the goal of this paper, we also follow the advice and  further apply our methods to expand  the full  CIFAR100 by 5 times for model training  from scratch. As shown in the following table, our GIF-DALLE leads to an 8.7% accuracy gain compared to  direct training  on the original CIFAR100  dataset. Such encouraging result further verifies the effectiveness of   our methods on larger-scale datasets, and we expect to apply our methods to   even larger datasets like ImageNet and other tasks in the future. We have provided this discussion in Appendix E.7 of the revised paper.
>
>
>
> |  CIFAR100           |    Accuracy    |
> | ---------- |:---------------:|
> | Original   dataset    |    70.9+/-0.6     |
> | 5x-expanded by GIF-MAE  |  77.0+/-0.3|
> | 5x-expanded by GIF-DALLE  | **79.6+/-0.3** |
>
>
> ---
> **Q2. Quality can be further improved with more theoretical analysis.**
>
>
> In this work, we aim to explore the empirical possibility of using recent generative models (*e.g.*, DALL-E2) for small dataset expansion, while  theoretical analysis is not the focus of this paper. Please note that rigid theoretical analysis for dataset expansion is complex and non-trivial, so here we  provide an exploratory idea to analyze the benefits of our dataset expansion   to generalization performance.
>
> Inspired by the work [A], we resort to the concept of $\delta$ cover    to analyze how the diversity of data influences  the generalization error bound. Specifically,  "a dataset $s$ is a  $\delta$ cover of a dataset $\hat{s}$" means a set of balls with radius $\delta$ centered at each sample of the dataset $s$ can cover the entire dataset $\hat{s}$. In our work, we  only consider the small target dataset and its true data distribution, so we follow the assumptions of  Theorem 1 in work [A] and extend it  to the version of the generalization error bound (Please refer to Appendix G for clearer formatting of the corollary).
>
> Let $A$ be a learning algorithm and $C$ be a constant.  Given a   training set $D=(x_i,y_i)$ for ${i \in [n]}$  with $n$ *i.i.d.* samples drawn from the true data distribution $P_Z$. If the training set $D$ is $\delta$ cover of the true  distribution $P_Z$, the hypothesis function is  $\lambda^{\eta}$-Lipschitz continuous, the loss function $\ell(x,y)$ is $\lambda^{\ell}$-Lipschitz continuous for all $y$ and bounded by $L$, and $\ell(x_i,y_i;A)=0$ for $\forall i\in [n]$, with the probability at least $1-\gamma$,   the generalization error bound satisfies  $|E_{x,y \sim P_Z}[\ell(x,y;A)]- \frac{1}{n}\sum_{i\in [n]}\ell(x_i,y_i;A)| \leq \delta (\lambda^{\ell}+\lambda^{\eta}LC)$.
>
> This corollary shows that the generalization error is bounded by the covering radius $\delta$. In real small-data applications, the data limitation issue leads $\delta$ to be very large and thus severely affects the generalization performance of the trained model.  More critically,   simply increasing the data number (*e.g.*, via data repeating) does not help the generalization since it does not   decrease the covering radius $\delta$. Rather than simply increasing the  number of samples, our proposed GIF framework  guides recent generative models (*e.g.*, DALL-E2) to synthesize informative and diversified new images for expanding the original small dataset. Therefore, the expanded dataset would have higher diversity of data, which helps to decrease the covering radius $\delta$ and thus improves the trained model's  generalization performance. In the revised paper, we have provided this exploratory theoretical analysis in Appendix G.
>
>
>
>
> [A] Active Learning for Convolutional Neural Networks: A Core-Set Approach. In ICLR, 2018.

---

### Official Review · Reviewer_cRYy · 2022-10-31

**Confidence:** 3
**Correctness:** 3
**Technical Novelty And Significance:** 2
**Empirical Novelty And Significance:** 2
**Recommendation:** 3

**Clarity, Quality, Novelty And Reproducibility:**

In terms of clarity, this manuscript is generally well-written. It could be better to include some details of the main algorithm, but the appendix helps me a lot enough to implement the approach.

In terms of reproducibility, all experiments have been conducted based on the publicly available checkpoints. And, the manuscript contains the hyperparameters and detailed experiments setups, so I believe that I wouldn’t be in trouble for reproducing the numbers in the tables.

In terms of quality and novelty, I have some major issues on the final performance of this method. The numbers in Table 1 are much worse than the linear evaluation and fine-tuning results in CLIP and ALIGN papers. This makes me wonder if this kind of dataset expansion based on a generative model is really better than the standard SSL approaches.

**Strength And Weaknesses:**

**Strengths**:

* Expanding the dataset based on a powerful decoder is a very intuitive and interesting direction to few-shot learning. In addition, this work proposes three criteria on how to create effective synthetic samples. And, these have been empirically validated in several experiments.
* The appendix helps me understand the details of the algorithm to reproduce the results. And, all experiments have been conducted on publicly available checkpoints, so the accessibility of this approach is quite high.

**Weaknesses**:

My biggest concern in this work is the low performance of the main experiments, shown in Table 1. Compared to the transfer learning or linear evaluation results from ALIGN and CLIP, the numbers in Table 1 aren’t convincing enough to argue that this kind of data expansion policy is practically useful than the standard protocol, i.e. training an encoder by SSL objectives then fine-tuning the encoder on small-scale datasets.

I summarize the performance gap between this method and CLIP and ALIGN below:

| Method | Caltech101 | Cars | Flowers | Pets |
| ---------- | -------------- | -------------- | -------------- | -------------- |
| GIF-MAE (proposed one) | 58.4 | 44.5 | 84.4 | 52.4 |
| GIF-DALLE2 (proposed one) | 63.0 | 53.1 | 88.2 | 66.4 |
| CLIP (ViT-B/32) - linear eval.    | 93.0  | 81.8  | 96.9 | 90.0 |
| CLIP (ViT-L/14, 334px) - linear eval.    | 96.0 | 91.5 | 99.2 | 95.1 |
| ALIGN - transfer learning | - | 96.1 | 99.7 | 96.2 |

ALIGN: Scaling Up Visual and Vision-Language Representation Learning With Noisy Text Supervision, https://arxiv.org/abs/2102.05918

**More questions**

Q1. In my understanding, the roles of the entropy maximization and diversity promotion seem to be very similar to each other. Could you elaborate a little bit more on the reason why both conditions are required instead of using the diversity promotion?

Q2. In table 1, GIF-DALLE consistently performs better than GIF-MAE, though the resolution of the synthetic samples based on GIF-MAE is much higher than the ones on GIF-DALLE (224px vs. 64px.), as described in Appendix C.1. It would be helpful to describe the reason why GIF-DALLE performs better than the MAE in the revised manuscript.

**Summary Of The Paper:**

This work improves the performance of low-shot learning by expanding the small-scale dataset by utilizing powerful generative models (i.e. decoders). To expand the dataset in a more meaningful way, this work proposes three conditions to ensure that the synthetic samples should be consistent with the real samples in zero-shot prediction perspective and diverse enough to improve the final performance of the models trained on the expanded dataset. Experiments on the proposed methods based on MAE and DALL-E 2 (without upscaling modules) show that this expansion policy indeed improves the classification performance in a low-shot setting.

**Summary Of The Review:**

I’m leaning towards rejection, since I’m not sure that the performance gain achieved by this work is really significant or not, compared to the simple recipes used in many SSL methods.

---

> ### Author Response · Authors · 2022-11-18
> **Response to Reviewer cRYy**
>
> Thanks a lot for the valuable comments, particularly for recognizing that our exploration of dataset expansion is intuitive and interesting.  We address the main concerns point by point as follows.
>
> ---
>
> **Q1. Concern about the performance of dataset expansion compared to the   models like CLIP.**
>
> This paper explores dataset expansion to handle real small-data scenarios, where   only a small-size dataset is available without any external large-scale  datasets **with similar image nature to the target dataset**. Therefore, pre-training  models with CLIP or other self-supervised methods  on large-scale **target** datasets  is inapplicable. In this work, we resort to publicly  available CLIP models for dataset expansion. Compared to directly transferring CLIP models, our  dataset expansion  is   a necessarily new paradigm for the following two key reasons.
>
> - **Our dataset expansion method has better applicability to  the scenarios of different image domains**. Although CLIP has strong  transfer performance on some natural image datasets, its transfer performance to other domains like medical image datasets is limited. Here, we further report the fine-tuning performance of CLIP-trained ResNet-50 model on three medical datasets. As shown in the following table,  transferring   the CLIP model   only leads to  limited performance gains, which are significantly worse than our dataset expansion. The reason is that,   when the pre-trained datasets are highly different from the target dataset, the pre-training and fine-tuning scheme does not significantly help performance [A]. In contrast, our GIF framework can generate images of similar nature as the target dataset for expansion, and thus   is more beneficial to real applications of various image domains.
>
>   [A] Transfusion: Understanding transfer learning for medical imaging. In NeurIPS, 2019.
>
>
> |  Method          |   PathMNIST   |    BreastMNIST   |    OrganSMNIST   |
> | ---------- |:---------------:| :---------------:| :---------------:|
> | Original  dataset     |    72.4 +/-0.7     |   55.8+/-1.3   |    76.3+/-0.4     |
> | Fine-tuned CLIP   |   78.4+/-0.9     |  67.2+/-2.4   |   78.9+/-0.1     |
> | GIF-MAE  |  82.0+/-0.7 |   73.3+/-1.3      |     80.6+/-0.6    |
> | GIF-DALLE (fine-tuned DALL-E)  | 84.4+/-0.3  |   76.6+/-1.4      |    80.5+/-0.2  |
>
>
>
>
> -  **Our dataset expansion  can provide expanded datasets ready for training various network architectures**. In some real application scenarios like mobile terminals, the supportable  model size  is very limited due to the constraints of   device hardware. However, the publicly available checkpoints provided by CLIP are only ResNet50, ViT-B/32 or even larger models, which may not  be allowed to use in those scenarios.  By contrast, the expanded dataset by our method can be directly used to train various model architectures  (cf. Table 2), and thus is more   applicable to the  application scenarios with hardware constraints. Also, once the  datasets are expanded, they  can   be released to the public to facilitate future studies.
>
>
> Please note that the goal of  this paper  is   **to show the huge potential of dataset expansion as a promising future  direction  instead of  completely resolving it**. We expect  that the performance of dataset expansion can be further improved in   future research.  We have further clarified this in the Appendix E.6 of the revised paper.
>
>
>
>
> ---
>
> **Q2. Are the roles of   entropy maximization and diversity promotion  similar to each other?**
>
> The two guidances play different roles. Entropy maximization  promotes   the informativeness of each generated image by increasing the prediction difficulty  over the  corresponding seed image, but this guidance cannot diversify different  latent features obtained from the same   image. By contrast, the guidance of  diversity promotion encourages the diversity of  various latent features of the same seed image, but it cannot  increase the informativeness of generated samples regarding  prediction  difficulty. Therefore, using the two guidances together leads  the expanded images to be more informative and diversified,  thus bringing higher performance improvement   in the ablation study of Table 8  (cf. Appendix E.2). We have added this clarification to   Appendix E.2 of the revised paper.
>
>
>
>
> ---
>
> **Q3. Why does GIF-DALLE perform  better  than GIF-MAE, when the generated images by GIF-DALLE have lower resolutsion than GIF-MAE?**
>
> The image resolutions are the same   during    training, since all   images are  resized to 224x224  for model training  (cf. Appendix C.3).  DALL-E2 can generate images with new content, and thus   brings more   information to the expanded dataset and leads to more significant  performance gains over MAE.
>
>
> Thanks again for your valuable comments. We welcome and are happy to discuss any further questions you may have.

---

### Author Response · Authors · 2022-11-19
**General Response**

We sincerely appreciate the precious time and effort of all reviewers in reviewing our paper and providing highly constructive feedback. Besides the response to each reviewer,  here we would like to further 1) thank reviewers for their recognition of our work and 2) highlight the major modifications in our revision:

**1) We are glad that the reviewers appreciate and recognize our novelty and contributions.**

* The explored dataset expansion is a **new, intuitive and interesting** direction [cRYy,DLM1]
* Applying generative diffusion models to help discriminative tasks is a natural and straightforward idea. This is a **pioneering work** in this area [DLM1]
* Using generative models for the low-data regime is an **intuitive** solution that shows promises but has so far been struggling to obtain significant success. The paper is **a step in that direction** [nKUi]
* The proposed disturbance is **simple and effective** [DLM1]
* This paper has achieved very **promising** results over many benchmarks. Such an improvement is **significant** [DLM1]
* The diverse set of experiments provides **good insights** [nKUi] and shows the effectiveness of the proposed method [sxmr]



**2) We make the following major modifications in our revised paper  (highlighted in blue).**

* We have further discussed why   dataset expansion is a necessarily new paradigm for real small-data applications,  compared to directly transferring CLIP models, in Appendix E.6  [cRYy,sxmr]
* We have further clarified the different effects of entropy maximization and diversity promotion for dataset expansion in Appendix E.2 [cRYy]
* We have added a new experiment on  the original CIFAR100 dataset in Appendix E.7, which   further demonstrates the effectiveness of our method in expanding larger-scale datasets [DLM1,sxmr]
* We have provided a new  theoretical analysis in Appendix G, which shows that our dataset expansion is beneficial to the trained model's generalization performance [DLM1]
* We have  added a new empirical analysis for infinite data augmentation in Appendix E.8, which further demonstrates the superiority of our  method [nKUi]
* We add a new discussion on picking related data from larger-scale datasets for  dataset expansion in Appendix E.9, which further shows the superiority of our  method [nKUi]
* We have further clarified the three guidance in Appendix C.1  and     the evaluation of dataset expansion in Appendix C.3  [sxmr]
* We   add  a new discussion on training  with only expanded images in Appendix E.10, which   further shows the effectiveness of our dataset expansion [sxmr]
*  We   have shown the superiority of our method over CutMix for dataset expansion in Appendix E.10  [sxmr]
*  We   have modified our tile to  "*Expanding Small-Scale Datasets with Guided Imagination*". [sxmr]

Based on the valuable suggestions and the above modifications,   the quality of this paper is further improved. Please note that **our work aims to show the huge potential of dataset expansion as a promising future direction, rather than completely resolving it**. We expect that the performance and computational efficiency of dataset expansion can be further improved in future research.  We welcome and are happy to discuss any remaining questions the reviewers may have.

---

### Decision · Program_Chairs · 2023-01-20

**Decision:**

Reject

**Justification For Why Not Higher Score:**

As pointed out by the reviewers cRYy and sxmr, the performance of the GIF approach is poorer than CLIP and ALIGN on some of the datasets that GIF was tested in. The paper proposed their approach as a generally applicable domain. However, that is not necessarily true. This paper needs to go through a substantial amount of changes and thus would benefit from another round of reviews.

**Justification For Why Not Lower Score:**

There is no lower score.

**Metareview: Summary, Strengths And Weaknesses:**

### Summary
This paper proposes the *Guided Imagination Framework* (GIF) that expands the datasets with images generated from strong generative model decoders like DALLE2 and CLIP encoders. This would be potentially be really helpful for small datasets.  The paper uses both MAE and DALLE2 models as the decoder for the classification experiments that it focuses on.

Below I will list some of the weaknesses and strengths pointed out by the reviewers for this paper:

### Strengths
- A simple and intuitive idea that paves the way for an interesting research direction.
- The results should be easy to reproduce given the level of details provided in the paper.
- The diverse set of experiments provides good insights into the method.
- The paper is well-written and easy to follow.

### Weaknesses
- Poor performance of the main experiments. The performance of the method is worse than CLIP and ALIGN methods as pointed out by the reviewers on some of the datasets like Caltech 101, Pets, Flowers and Cars datasets.
- The datasets MAE and DALLE-2 are trained on my cover some of the datasets the GIF were evaluated on which would be problematic.
- GIF-DALLE's performance is much worse than the CLIP approach.
- Requirement for the heavy computations

**Decision:** The proposed idea is very interesting and the authors did a really good job answering the concerns raised by the reviewers. However, reviewer cRYy and sxmr both had serious concerns about the experiments and the paper. The proposed approach's performance was significantly worse than the approaches like CLIP and ALIGN. The reviewers suggested that the proposed GIF approach is more general in a way that it might provide improvements on tasks or datasets where CLIP/ALIGN may not be applicable, for example on medical image domain, because of the domain gap between the dataset that they were trained on vs evaluated on whereas GIF does not necessarily need to suffer from that. The authors tried to addressed those concerns during the rebuttal. However, the authors need to do substantial amount of changes on the paper to address these concerns. Thus, I think the paper might benefit from another round of reviews. Thus I recommend the authors to resubmit this work to another venue, after addressing the concerns raised by the reviewers.